# TASK REPRESENTATIONAL DYNAMICS FOR COMPOSITIONAL GENERALIZATION DURING CONTEXT-DEPENDENT COGNITIVE TASKS

## ABSTRACT

The ability to generalize compositionally is central to intelligent behavior. While recent work shows that networks can generalize compositionally under certain conditions, many studies focus on simple compositional tasks, such as those that are purely linguistic (or unimodal), or those with no temporal structure. Here we investigate how representational dynamics shape compositional generalization in recurrent neural network (RNN) models during cognitive tasks with evolving temporal structure, providing insights into neural computation during flexible reasoning. We trained RNNs on the Concrete Permuted Rules (C-PRO) task, a cognitive compositional task established for humans that requires integration of information across task phases. We assessed how different learning regimes induced generalization and representational dynamics. We systematically varied model initializations to generate RNNs that exhibited a wide range of compositional generalization performance, ranging from 38% to 90%. Analysis of high-performing models revealed nontrivial temporal dynamics of task representations, highlighting the importance of selectively engaging the right features at the appropriate task phase for generalization. Our findings reveal that successful compositional generalization requires the orchestration of structured intermediate representations that are dynamically composed, resulting in complex, feature-specific representational dynamics – providing testable principles for how neural systems enable flexible reasoning.

## 1 INTRODUCTION

Understanding the mechanisms underlying compositional generalization requires examining how networks organize their internal representations during learning and task performance. Recent work has established that neural networks can achieve compositional generalization when equipped with useful and reusable representations (Yang et al., 2019; Ito et al., 2022; Johnston & Fusi, 2023; Driscoll et al., 2024). Other theoretical work has identified distinct learning regimes, with "rich" learning leading to structured representations that likely enhance compositional generalization (Lippl & Stachenfeld, 2024), while "lazy" models learn input-output mappings by projecting input features to a random, high-dimensional space, similar to reservoir computing (Chizat et al., 2019). However, these prior studies have primarily examined the structure of representations without considering cognitive tasks that require the temporal orchestration of information, and have typically focused on simple categorization or context-dependent tasks with limited manipulation of task features. This limits the insights from prior studies to more practical settings, where decisions in realistic cognitive tasks unfold over time and require dynamic integration of evolving information. Thus, a computational understanding of how representational dynamics and learning regimes jointly influence compositional generalization remains limited.

When computational experiments are paired with well-designed cognitive tasks, they can help uncover the mechanisms underlying specific cognitive processes. Here, we study computational models performing the Concrete Permuted Rules Operation (C-PRO) task, a task commonly used to probe compositionality in humans. Successful C-PRO performance requires 1) composability across diverse feature domains, and 2) dynamic integration of these features within and across distinct temporal phases. Characterizing how models perform this task extends prior work by tracking how

representations of context, stimuli, and response information evolve to support compositional generalization.

A promising approach for understanding these dynamics is through the lens of neural representational dimensionality and decoding (Badre et al., 2021). Studies have shown that high-dimensional, decodable representations during stimulus integration improve task performance (Rigotti et al., 2013; Fusi et al., 2016; Kikumoto & Mayr, 2020), while compressed, lower-dimensional abstract representations enable generalization (Bernardi et al., 2020; Flesch et al., 2022; Ito & Murray, 2023; Chakravarthula et al., 2025). These findings may appear contradictory, but prior studies primarily characterized representational geometry during isolated task phases (e.g., instruction period vs. stimulus presentation). We adjudicate these differences by tracking the evolution of dimensionality and decodability across all phases, assessing their impact on compositional generalization.

We trained RNNs on the C-PRO task to examine how rich vs. lazy learning regimes affect representational dynamics and compositional generalization. Consistent with prior work, rich learning led to significantly better generalization. However, while previous work has emphasized that rich learning tends to produce low-dimensional, structured representations that support generalization (Flesch et al., 2022; Ito & Murray, 2023; Farrell et al., 2023; Liu et al., 2023), our findings revealed that examining the geometry of feature-specific representations – specifically stimulus, context, and response information – provides critical insights into compositional generalization in tasks with dynamic temporal structure. Specifically, we found that successful generalization emerged through coordinated temporal dynamics across feature representations. High-performing rich networks displayed feature representations that expanded and compressed throughout different task phases, while lazy networks showed relatively simplistic temporal dynamics. Interestingly, the dynamic coordination observed in rich networks enabled the formation of contingency representations that enabled the formation of early and efficient decisions. Together, these findings demonstrate that compositional generalization in temporally structured tasks depends on the development of structured, feature-specific temporal dynamics.

## 2 EXPERIMENTAL DESIGN

### 2.1 THE CONCRETE PERMUTED RULES OPERATION TASK

The C-PRO paradigm studies context-dependent compositionality, executive control, and task generalization in humans (Ito et al., 2017; Cole et al., 2013). C-PRO composes task rules from three domains (logical decision, sensory semantic, and motor response) to generate up to 64 task contexts (Fig. 1B-C). The sensory rule indicates which stimulus feature to attend (*Is it red? Is it vertical? Is it high pitch? Is it periodic?*); the logic rule specifies the Boolean operation (AND/NAND/OR/NOR) applied to two sequentially presented stimuli; and the motor rule specifies the required action (button press with a specific finger, corresponding to four output channels in our computational implementation). While the human C-PRO task involves naturalistic multimodal stimuli, we remapped these conditions and sensory modalities to multi-hot encoded stimulus input vectors for computational experiments. Successful performance requires flexibly gating the relevant sensory dimension and assigning motor responses according to the logical evaluation of sensory information.

To study the evolution and dynamics of task representations, we imposed a temporal structure mimicking human experiments (Fig. 1B). The structure consists of six phases: 1) task encoding where the 3-rule context is presented (1 timepoint, Rule); 2) first stimulus presentation (2 timepoints, early and late, Stim1_e and Stim1_l); 3) first delay period for working memory and integration (2 timepoints, Dly1_e and Dly1_l); 4) second stimulus presentation (2 timepoints, Stim2_e and Stim2_l); 5) second delay period (2 timepoints, Dly2_e and Dly2_l); 6) motor response window (Resp). In our computational model, task context input remains active throughout, though we also examine variants where context is limited to the first timepoint (Appendix Fig. A4).

### 2.2 TASK DIMENSIONS: CONTEXT, STIMULI, AND RESPONSE FEATURES

The C-PRO contains three task dimensions: task context, stimuli, and response features (Fig. 1C).

**Sensory Stimuli.** In the human experiment, stimuli were defined by four feature categories (two visual and two auditory categories): color (red/blue), orientation (vertical/horizontal visual line),

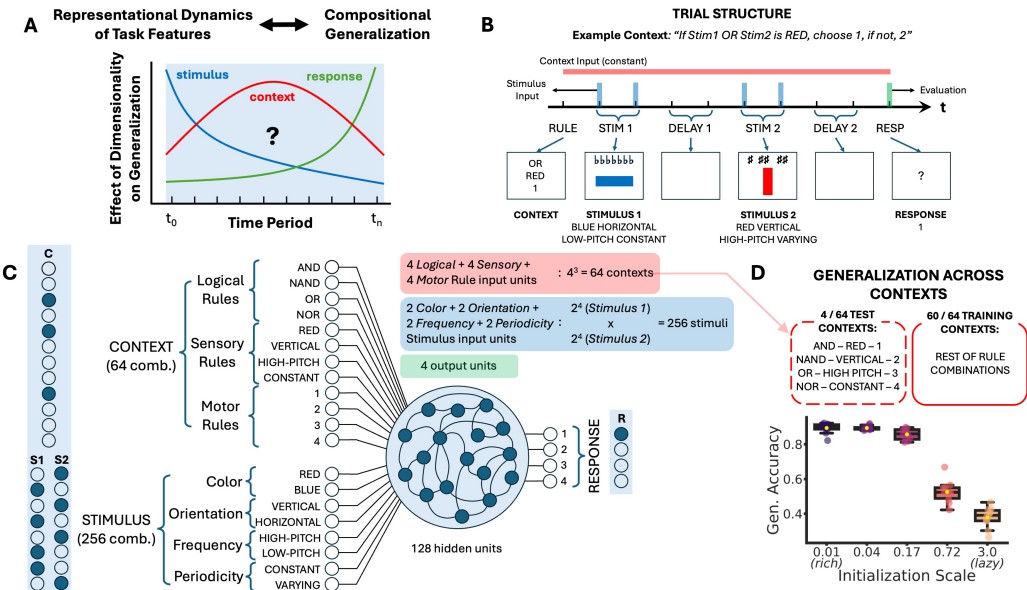

Figure 1: **Experimental overview.** We study how representational dynamics contribute to generalization in a compositional task by manipulating model initializations. **A)** We examined how dynamics of task feature representations contribute to generalization. **B)** We modeled six temporal phases within the C-PRO task: Rule encoding, Stimulus 1, Delay 1, Stimulus 2, Delay 2, and Response. Task rules compositionally combined logical operations (AND/OR/NOR/NAND), sensory features (RED/VERTICAL), and motor responses (see section 2.1). **C)** One-hot encoded inputs are projected to a 128-unit RNN initialized with either high-norm (lazy learning) or low-norm (rich learning) weights. **D)** Rich models achieved 90% generalization accuracy vs. 38% for lazy models.

frequency (high/low auditory pitch), periodicity (high/low period) (Fig. 1B). To model this computationally, we encoded each of these stimulus features as their own separate embedding dimension, represented as a multi-hot encoding. For simplicity, we defined the binary stimulus vector $s \in \{0, 1\}^2$ for each of the four stimulus dimensions (color, orientation, frequency and periodicity). A given stimulus activated one of the two features per each category (activating four category units in total). Furthermore, the C-PRO task presented two stimuli across task phases. This resulted in in $2^4 \times 2^4 = 256$ possible stimulus combinations across the two stimuli presentations trial.

**Context.** Task context is composed from three rule domains (sensory, logic, and motor/output). Each domain has four specific rules. **Sensory rules** – RED, VERTICAL, HIGH-PITCH, CONSTANT – specify which stimulus feature to attend to or gate. These correspond to a specific color, orientation, frequency, or periodicity dimensions respectively. **Logical rules** – AND, NAND, OR, NOR – determine how to logically evaluate the gated sensory dimension across the two stimulus phases (e.g., AND × RED → "*Are both stimuli red?*". **Motor rules** – left index, left middle, right index, right middle finger responses – specify the finger to respond with if the answer to the sensory-logical evaluation is TRUE. If the evaluation is FALSE, e.g., if the logic and sensory rules are AND and RED, but one of the stimuli is blue, then the other finger *on the same hand* is used. Computationally, a rule from each rule domain is represented as a one-hot vector $r_x \in \{0, 1\}^4$, for $r_{\text{logic}}, r_{\text{sensory}}, r_{\text{motor}}$. The full task context vector is specified by concatenating the vectors across each rule domain, i.e., $r_{\text{context}} = [r_{\text{sensory}}, r_{\text{logic}}, r_{\text{motor}}] \in \{0, 1\}^{12}$. Permuting four rules across three rule domains results in 64 possible unique contexts ($4 \times 4 \times 4$ combinations):

**Motor/output vector.** Computationally, we map each finger to an embedding dimension of a one-hot vector $o \in \{0, 1\}^4$.

## 2.3 EVALUATING GENERALIZATION

Our primary goal was to assess the influence of training regime and learned representational dynamics on compositional generalization. This required a subset of tasks where we 1) measured generalization and 2) performed mechanistic analyses. Thus, model training was performed on 60/64 task contexts. Generalization and mechanistic analyses were carried out on the remaining 4 contexts. The training set was chosen such that each individual rule was equally represented across the 60 training contexts. Generalization testing was then evaluated on the four remaining contexts, where each context was tested across all 256 unique trials (e.g., 256 unique stimulus combinations). Training regime was manipulated through scaling the norm of the weight initialization (see Appendix A.3 for further details).

## 2.4 MODEL AND DIMENSIONALITY METRIC DETAILS

We implemented an RNN architecture with 128 hidden units, 20 input units (12 context + 8 stimulus), and 4 output units. The model used ReLU non-linearity with layer normalization applied before the activation function. To manipulate learning regimes, we scaled Xavier initialization weights for the recurrent connections. We tested 5 logarithmically-spaced initialization values: $\{0.01, 0.04, 0.17, 0.72, 3.0\}$. For each initialization scale, we trained 10 networks with different random seeds, yielding 50 networks total. Training proceeded for 1000 epochs using vanilla stochastic gradient descent with a batch size of 128 and learning rate of 0.01. Each epoch comprised 15,360 training examples (60 tasks × 256 stimuli per task). We include additional experiments varying model size and optimizer in the Appendix.

We characterized representational dimensionality at each time point separately. Representational dimensionality measures how many dimensions are required to capture meaningful variation in neural representations. If the 64 task contexts were stored as unique contexts (i.e., each orthogonal to each other), they would span a 64-dimensional space. However, if some contexts share structure (e.g., through overlapping rules), they can be represented in a lower-dimensional subspace. We quantified this using the participation ratio:

$$\text{PR} = \frac{(\sum_{i=1}^{s} \sigma_i)^2}{\sum_{i=1}^{s} \sigma_i^2}$$

where $\sigma_i$ represents the singular values from SVD decomposition of neural activity matrices. Higher values indicate more distributed, high-dimensional representations. We measured this metric globally across all trials and separately for each feature (stimulus, context, response, contingency) by averaging activity across the other features.

## 3 RESULTS

Networks exhibited a striking dependence on initialization scale for compositional generalization. Rich networks (smallest initialization scale) achieved 89 (±2.9 SD)% mean accuracy across seeds, demonstrating robust compositional generalization (see Appendix Fig. A7 for detailed breakdown). In stark contrast, lazy networks (largest initialization scale) achieved only 38 (±5.7 SD)% accuracy, barely above chance level (25%) (Fig. 1D), despite near-perfect training accuracy on 60 training contexts (Fig. A1). This contrast in generalization performance motivated our subsequent analyses of the internal representational dynamics underlying successful compositional generalization.

## 3.1 REPRESENTATIONAL DYNAMICS ACROSS TASK PHASES

We quantified dimensionality of feature-specific representations (stimulus, context, response) and global dimensionality to assess representational dynamics across models (Fig. 2). Lazy networks exhibited stereotyped dynamics with dimensionality generally increasing across task phases (Fig. 2D-G). In contrast, rich networks showed feature-specific heterogeneous dynamics: stimulus dimensionality increased during stimulus presentations then compressed after encoding both stimuli (Fig. 2D); context dimensionality fluctuated with suppression during stimulus periods and re-emergence during delays (Fig. 2E); response dimensionality rose earlier in rich networks (Fig. 2F); global

dimensionality showed marked compression during response (Fig. 2G). These dynamics were consistent across different model sizes (Appendix Fig. A2) and optimizers (Appendix Fig. A3).

To validate whether dimensionality changes reflected information content versus noise, we performed cross-validated decoding of task features (Fig. 2H-K). Decoding confirmed that rich networks' stimulus dimensionality expansion tracked increased decodable information (Fig. 2H). Interestingly, however, high feature dimensionality did not always imply high decodability; in fact, lower context dimensionality was associated with greater decoding accuracy, suggesting efficient compression of context information aided decodability and generalization performance (Fig. 2I). Furthermore, response information emerged independently of dimensionality changes in rich networks (Fig. 2J). Critically, context-relevant stimulus decoding revealed that rich networks selectively maintained task-relevant dimensions while irrelevant features decayed during delays, demonstrating active filtering rather than passive forgetting (Fig. 2K). Complementarily, subspace angle analysis confirmed rich networks transiently decoupled stimulus and context subspaces during delay periods (Appendix Fig. A6). We further quantified the cross-context representational invariance for each task feature in rich and lazy networks across time (Appendix Fig. A5), as well as a supporting analysis using RSA rather than decoding (Appendix Fig. A8). These distinct representational dynamics between rich and lazy networks motivated our subsequent analysis on how these representational patterns directly support compositional generalization.

## 3.2 REPRESENTATIONAL DYNAMICS FOR C-PRO GENERALIZATION

To examine how temporal variations in representational dynamics influenced generalization, we analyzed how task features across different phases contributed to generalization. This was done by performing regression analyses to predict how representational dimensionality (independent variable) at each task phase contributed to generalization performance (dependent variable), with each of the 50 models (5 initialization scales × 10 seeds) serving as a sample. This enabled us to infer whether higher or lower representational dimensionality affected generalization across task phases. We first analyzed how global dimensionality contributed to generalization across task phases, followed by investigating how the dimensionality of task-specific features influenced generalization (Fig. 3).

### 3.2.1 GLOBAL DIMENSIONALITY DYNAMICS AND GENERALIZATION

Global dimensionality – which captures the overall dimensionality of trial-wise neural activity patterns – revealed temporally specific relationships between dimensionality and generalization (Fig. 3A). We found that during the rule encoding phase (`Rule`), low dimensionality supported generalization. Following rule encoding, the initial stimulus presentation period (`Stim1_e`, `Stim1_l`, `Dly1_e`) exhibited the opposite pattern, where high dimensionality supported generalization. Late task periods (`Dly1_l → Resp`) showed that increasingly lower dimensionality improved generalization. These patterns reconcile contrasting neuroscience findings showing high-dimensional representations support stimulus integration (Rigotti et al., 2013; Kikumoto et al., 2024) while low-dimensional representations aid task encoding (Bernardi et al., 2020), revealing task-phase dependent dimensionality requirements. Global dimensionality's explanatory power ($R^2$) showed temporal progression: low ($R^2 < 30\%$) until `Dly1_e`, then increasing to $>90\%$ by trial end (Fig. 3B), indicating late-stage representations primarily account for generalization variance.

### 3.2.2 FEATURE-SPECIFIC DIMENSIONALITY DYNAMICS AND GENERALIZATION

Next, we characterized how representational dynamics of specific task features contributed to generalization by quantifying feature-specific dimensionality (stimulus, context, response) across all model initializations and using these as regressors predicting generalization performance. Decomposing global dimensionality into feature-specific components explained significantly more variance ($R^2 > 90\%$ from `Stim1_e` through `Resp`; Fig. 3F) than global dimensionality alone.

Context dimensionality showed that improved generalization was associated with consistently low-dimensional representations (Fig. 3D), consistent with prior work on abstract rule representations supporting cross-context generalization (Bernardi et al., 2020; Ito et al., 2022). Decoding analysis confirmed that low context dimensionality corresponded to higher decoding accuracy (Fig. 2I).

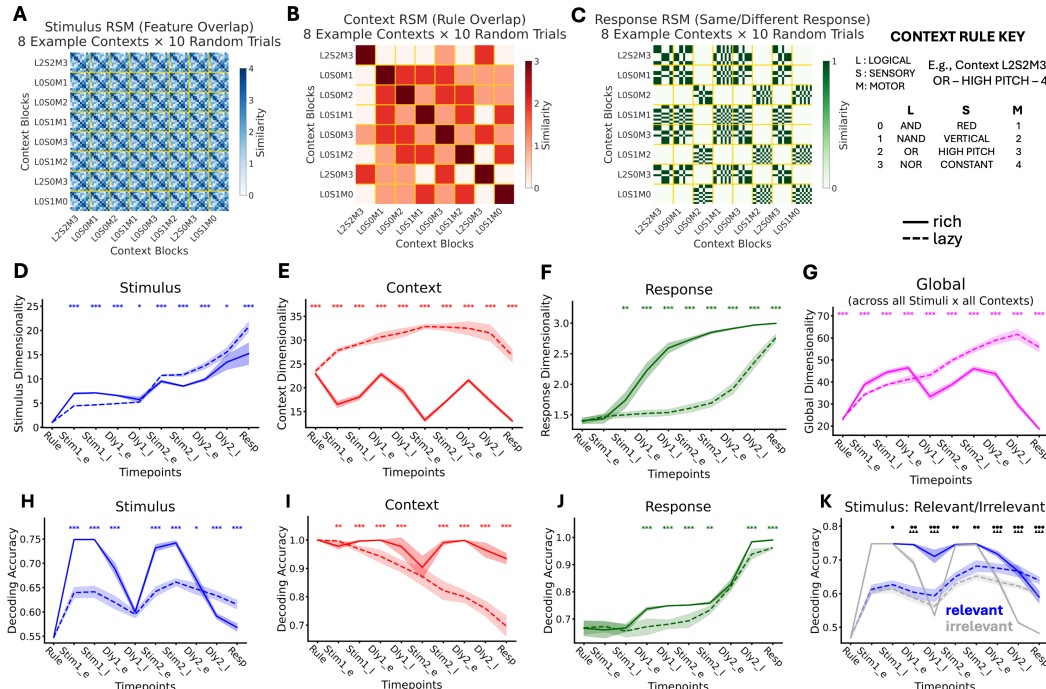

Figure 2: **Feature-specific and global representational dynamics.** We illustrate the ground truth feature similarity matrices for **A)** stimulus features, **B)** context features, and **C)** response features by computing the cosine similarity of each feature vector (either the ground truth input or output vector) across sample trials. **D-F)** Dimensionality dynamics of task-specific features in rich versus lazy networks for **D)** stimulus, **E)** context, and **F)** response. **G)** Global dimensionality measured across all 64 contexts × 256 stimulus combinations. **H-J)** Decoding traces using 10-fold cross-validation with distance-based classifiers. **H)** Rich models showed positive coupling between stimulus dimensionality and decodability. **I)** Context decodability and dimensionality show opposite relationships between regimes – rich models achieve higher decodability with lower dimensionality, while lazy models do not. **J)** Response dimensionality rises earlier in rich models, but decodable information emerges independently of dimensionality changes. **K)** Context-relevant versus -irrelevant stimulus decoding. Rich models preferentially gate and maintain relevant stimulus information (active selection); lazy models show minimal differentiation. Asterisks show FDR-corrected rich-versus-lazy comparisons (paired t-tests): * $p < 0.05$, ** $p < 0.01$, *** $p < 0.001$. Panel K uses triangles (rich) and circles (lazy) for relevant versus irrelevant stimulus comparisons.

Stimulus dimensionality revealed dynamic temporal strategies. High dimensionality during `Stim1_e` strongly predicted generalization (Fig. 3C), with decoding confirming dimensionality expansion tracked information content (Fig. 2H).

Response dimensionality showed high-dimensional encoding benefited generalization during rule encoding (Fig. 3E). Decoding showed response information emerged independently of dimensionality changes (Fig. 2J), with rich networks showing an earlier rise in dimensionality than lazy networks.

### 3.3 THE EFFECT OF TASK CLOSURE AND INTEGRATION ON REPRESENTATIONAL DYNAMICS

The C-PRO task structure integrates task context information with two sequentially presented stimuli. However, when studying the contributions of stimulus-related dimensionality on generalization, we observed that the first stimulus had an out-sized effect on generalization, compared to the second stimulus (Fig. 3C). This effect is likely due to the structure of the C-PRO task, where certain context and stimulus combinations enable early decision-making after the first stimulus. For example, if the task context was `AND - RED - LEFT MIDDLE`, corresponding to the instruction "*If Stim1*

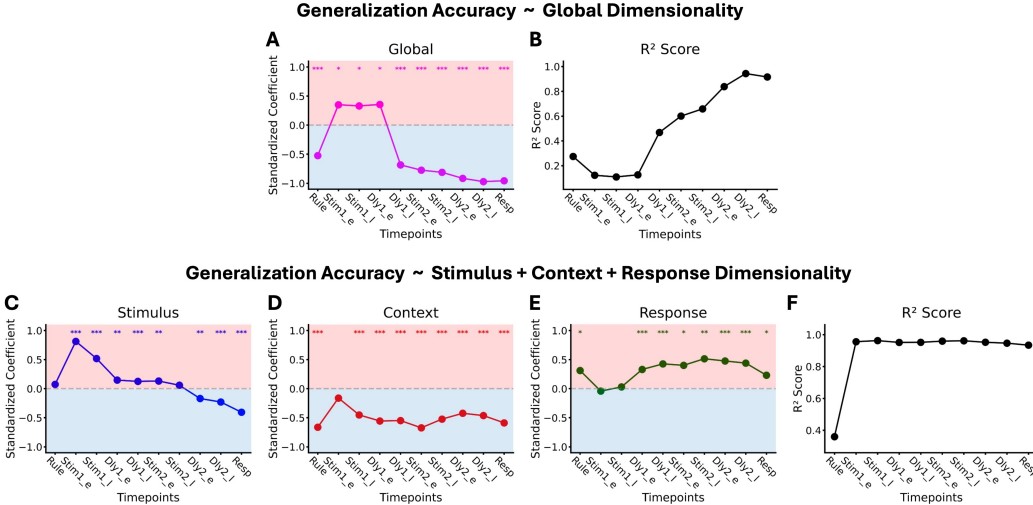

Figure 3: **Representational dynamics supporting compositional generalization.** Regression coefficients of representational dimensionality (global, stimulus, context, response) predict generalization performance across initializations. **A)** Global dimensionality coefficients: negative during rule encoding (low-dimensional improves generalization), positive during stimulus processing (high-dimensional improves), strongly negative near response (low-dimensional predicts best performance). **B)** $R^2$ for global dimensionality: low explanatory power during early phases rises to >90% by `Dly2_l`, showing late-stage low-dimensional representations primarily account for generalization variation. **C-E)** Multiple regression isolating feature-specific dimensionality contributions. **C)** Stimulus: high-dimensional at `Stim1` transitioning to low-dimensional by `Dly2` predicts better generalization. **D)** Context: consistently low-dimensional predicts better generalization. **E)** Response: sustained high-dimensional after `Stim1` predicts better generalization. **F)** Feature-specific $R^2$ exceeds 90% from `Stim1_e` through `Resp`, demonstrating that isolating task features better captures how representational dynamics support compositional generalization than global dimensionality alone. Asterisks show FDR-corrected comparisons of coefficients (t-test against zero): * p < 0.05, ** p < 0.01, *** p < 0.001.

*and Stim2 is red, press your left middle finger*", and the first stimulus was blue, then the response could be prepared after the first stimulus (Fig. 4A). This task effect has previously been described as "closure", and leverages contingency representations – representations that enable efficient mappings to future responses/decisions (Ehrlich & Murray, 2022). We refer to trials that do not exhibit closure – tasks that require both stimuli to form the correct response – as "integration" (Fig. 4A). In this section, we characterize the representational dynamics of high-performing models during closure and integration trials. Clear dissociation of representational dynamics across these two conditions provides strong evidence that these high-performing models leverage these highly efficient contingency representations that are common for human working memory and planning (Ehrlich & Murray, 2022).

### 3.3.1 CONTINGENCY REPRESENTATION DYNAMICS

Contingency representations correspond to Boolean evaluations of sensory and logic rules (e.g., *Are both stimuli red?*), providing intermediate representations that map efficiently to motor responses (*If True, respond with X*). Rich networks showed decreased contingency dimensionality during delay periods (Fig. 4B), corresponding to high decoding accuracy during these same periods (Fig. 4C), indicating accurate encoding of the task contingency. In contrast, lazy networks showed no such contingency representations (`Stim1_l → Dly1_e`, `Stim2_l → Dly2_e`: AnovaRM interactions p<0.0001). Strikingly, when evaluating closure and integration trials separately, we found that rich models achieved perfect decoding accuracy after `Stim1` in closure trials, while integration trials reached equivalent accuracy only after `Stim2` (Fig. 4D). This suggests that rich networks accurately

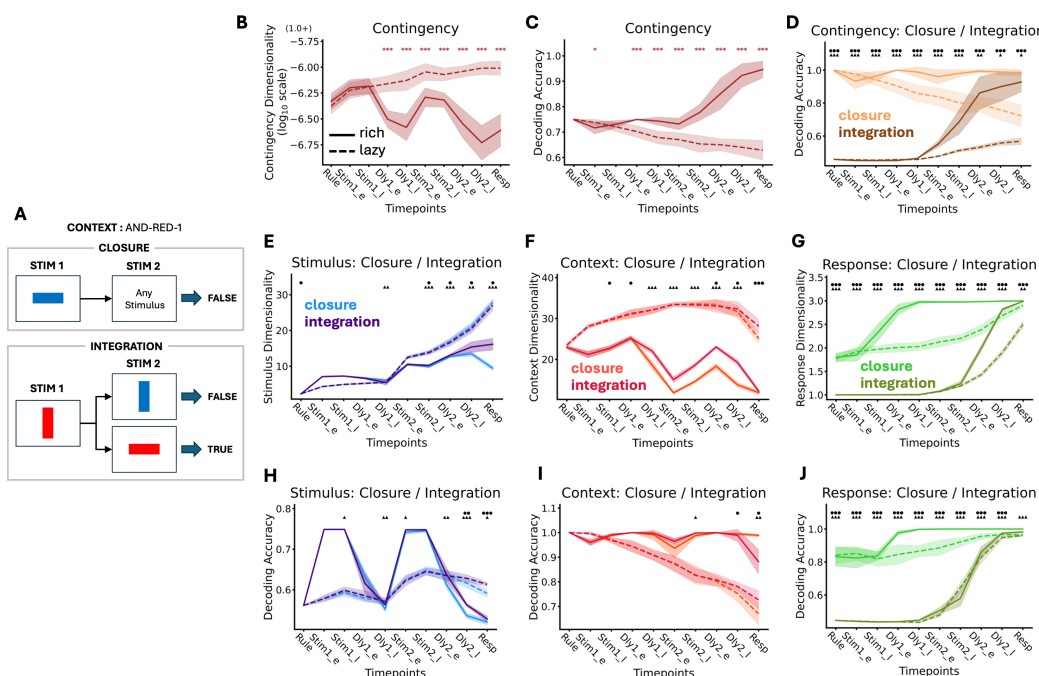

Figure 4: **High-performing models exhibit contingency representations that enable efficient early decision-making. A)** CPRO task structure enables early decisions after Stim1 in "closure" trials (logical rule resolvable immediately), but not "integration" trials (requiring both stimuli). In particular, these "early decisions" can be captured through contingency representations that encode intermediate information about the task (e.g., "Are both stimuli red?" → TRUE/FALSE. **B)** Dimensionality of contingency representations. **C)** Decoding analyses of contingency representations illustrated that contingency representations tended to emerge later in the trials. **D)** However, when analyzing closure and integration trials separately, decoding contingency representations revealed that rich models achieved perfect decodability as soon as decisions can be theoretically formed (after Stim1 in closure trials; after Stim2 in integration trials). Feature-specific dimensionality for closure versus integration trials for **E)** stimulus features, **F)** context features, and **G)** response features. Decoding traces for closure versus integration trials for **H)** stimulus features, **I)** context features, and **J)** response features. Asterisks show FDR-corrected rich-versus-lazy comparisons (paired t-tests): * p < 0.05, ** p < 0.01, *** p < 0.001. Panels D-J use triangles (rich) and circles (lazy) for closure versus integration comparisons.

encode intermediate contingency representations that would allow the early formation of response information. We next examine how contingency representations influence dynamics during closure versus integration trials across all task features.

### 3.3.2 REPRESENTATIONAL DYNAMICS OF CLOSURE AND INTEGRATION TRIALS

Contingency representations influenced feature-specific dynamics differently during closure (decision after Stim1) versus integration (requiring both stimuli) trials (Fig. 4E-J). Stimulus dimensionality and decoding both showed lower values during the presentation of the second stimulus when comparing closure versus integration trials (Fig. 4E,H), reflecting reduced stimulus processing when contingencies are already formed. Context dimensionality decreased in closure trials from Dly1_e onwards while decoding remained at ceiling (Fig. 4F,I). Response dynamics revealed the starkest contrast, and their trajectory best tracks with the emergence of decodable contingency representations: both dimensionality and decoding of response information rose immediately after Stim1 in closure trials, but only after Stim2 in integration trials (Fig. 4G,J). Together, these results provide

clear evidence that early response information in closure trials is enabled by the early formation of intermediate contingency representations.

# 4 DISCUSSION AND CONCLUSION

Our findings revealed that successful compositional generalization emerges from structured representational dynamics that enable systematic interactions between context, stimuli, and response task features. In particular, we found that high-performing rich networks develop three key signatures: strategic dimensionality modulation (low-dimensional context encoding, high-dimensional stimulus processing with selective compression, and flexible response timing; Fig. 2), temporal decoupling between stimulus and context subspaces during delay periods (Fig. A6), and efficient formation of intermediate contingency representations that enable early decision strategies (Fig. 4, A5). These dynamics distinguish improved generalization in rich networks from lazy networks, which exhibited non-specific temporal dynamics across features, and failed to form reliable contingency representations critical for the correct response.

## 4.1 RELATED WORK

**Representational dynamics in models for computational neuroscience.** Prior work has demonstrated that gradient descent optimization drives RNNs toward dimensionality compression in simple tasks, with greater compression supporting better generalization (Farrell et al., 2022). Other work has shown that networks trained on multiple tasks develop recurring computational motifs that enable compositional computation (Driscoll et al., 2024; Yang et al., 2019; Kay et al., 2024; Johnston & Fusi, 2023; Ito et al., 2022). Our findings complement these results by revealing how feature-specific representational dynamics – rather than global compression alone – enable compositional generalization through stereotyped temporal dynamics of stimulus, context, and response representations.

**Rich and lazy learning regimes.** Our work extends the feature learning vs. function fitting trade-off in the rich/lazy spectrum (Chizat et al., 2019; Flesch et al., 2022; Tong & Pehlevan, 2025; Lippl & Stachenfeld, 2024; Liu et al., 2023). While prior work focused primarily on static tasks (e.g., see Farrell et al. (2023) for review) or learning dynamics *during optimization* (Chou et al., 2025; Dominé et al., 2024; Kunin et al., 2024; Liu et al., 2023), we examine how task feature representations evolve *after* the model has been optimized. Critically, these systematic representational dynamics predict generalization performance and reveal efficient, intermediary contingency representations that enable early decision making (Ehrlich & Murray, 2022).

**Relation to empirical neuroscience and cognitive science.** Our findings help reconcile contrasting views on how representational dimensionality supports task performance in empirical neuroscience. Some studies emphasize high-dimensional representations during stimulus and delay periods for integrating stimuli with task context (Kikumoto & Mayr, 2020; Kikumoto et al., 2024; Fusi et al., 2016; Rigotti et al., 2013), while others highlight low-dimensional context representations as crucial for generalization through "abstract or disentangled representations" (Bernardi et al., 2020; Flesch et al., 2022; Courellis et al., 2024). Our results revealed that these seemingly contrasting properties reflect distinct computational phases: networks compress context representations during rule encoding, then expand stimulus representations during integration. Crucially, we demonstrate that feature-specific temporal coordination – not global dimensionality patterns – predicts generalization success.

**Compositional generalization in the machine learning literature.** Substantial effort in machine learning has developed benchmarks for compositional generalization (Keysers et al., 2020; Lake & Baroni, 2018; Hupkes et al., 2020; Johnson et al., 2017; Ruis et al., 2020; Kim & Linzen, 2020; Delétang et al., 2022), characterizing fundamental limitations of neural networks (Dziri et al., 2023; Kim et al., 2022; Lake & Baroni, 2018) and approaches for improving generalization in lazy regimes (Canatar et al., 2021; Abbe et al., 2024). While specialized architectures or training regimes can improve compositional performance (Lake & Baroni, 2023; Csordás et al., 2022; Sinha et al., 2024; Zhou et al., 2023; Ontanon et al., 2021; Kazemnejad et al., 2023), these benchmarks typically study unimodal (e.g., linguistic) or static settings without the temporal integration demands of real-world cognitive tasks. Our work differs from traditional compositionality studies in ML by focusing our

efforts on temporally-structured compositional paradigms established in the human literature to address compositionality in cognitive science. While our experiments focus on a single task and architecture (RNNs), future work should assess generalizability across diverse cognitive demands, network architectures, and human data.

## 4.2 LIMITATIONS

Our study highlights the critical role of structured temporal representations in enabling robust compositional generalization in RNNs. Nonetheless, several limitations remain in the present work. First, while our implementation of the C-PRO task preserves its core cognitive operations – compositional reasoning, working memory integration, and contingency formation – we used symbolic input encodings (e.g., multi-hot encodings of binary features) rather than naturalistic stimuli used in the original human task design. While this limits direct comparisons to the naturalistic sensory processes involved in human cognition, this allowed us to isolate the cognitive processes of interest (compositional reasoning). Nevertheless, it will be interesting for future work to explore the temporal evolution of representations in compositional tasks that use naturalistic stimuli. Second, we studied the contribution of representational dynamics to compositional generalization by manipulating RNN weight initialization. While prior work has demonstrated that weight initialization can strongly influence generalization (Chizat et al., 2019; Farrell et al., 2023), it is possible that there are other model manipulations – as well as model architectures and optimization protocols – that induce different relationships between representational dynamics and generalization that future work can explore. Finally, we limited our investigation to the C-PRO task. While the C-PRO task has a highly stereotyped task configuration that is commonly-used within neuroscience and cognitive science (i.e., context $\rightarrow$ stimulus $\rightarrow$ response), it will be important to explore compositional task designs in future work.

## 4.3 CONCLUSION

Successful compositional generalization in temporal cognitive tasks requires learning representational dynamics that enable systematic coordination across task features. Networks with different initialization scales achieve equivalent training accuracy (>99% on trained contexts) yet show dramatically different generalization (90% vs. 38% on held-out contexts), allowing us to isolate which temporal dynamics distinguish compositional success from failure. Through convergent evidence from dimensionality, decoding, and representation analyses, we demonstrated that rich networks coordinate temporal transformations across stimulus, context, and response features: context dimensionality modulates strategically across task phases including increases during working memory delays, stimulus representations expand during presentation periods then selectively compress, and response representations emerge with timing that reflects when decisions can be formed. Critically, feature-specific dimensionalities predict $> 90\%$ of generalization variance from early task phases, while global dimensionality achieves comparable predictive power only at trial end. This demonstrates that the relevant compositional structure manifests in coordinated feature dynamics. These dynamics also enable computational efficiency, as rich networks exploit task structure that enable early decisions to form when possible. Together, our findings establish that successful compositional generalization emerges from temporally coordinated transformations across feature spaces that enable systematic interactions between task features throughout processing phases.

## 5 REPRODUCIBILITY STATEMENT

All code and environment needed to reproduce our models, results, and figures are included in the supplementary materials.

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

# A    APPENDIX

## A.1    LARGE LANGUAGE MODEL USAGE

Claude Sonnet 4 was used to refine grammar, improve sentence structure, and enhance readability. All substantive content and ideas remain original to the authors.

## A.2    TASK DESIGN AND EXPERIMENTAL SETUP

### A.2.1    CPRO TASK STRUCTURE

We operationalized the Concrete Permuted Rules Operation (C-PRO) to study compositional decision-making across a temporally structured task. The task rule/context was composed of three distinct rule components: logical operations, sensory contexts, and motor responses. Each experimental trial consisted of two sequentially presented stimuli (Stim1 and Stim2) followed by a response period, interleaved with delay periods. The task structure incorporated 20-dimensional input vectors with 4 stimulus dimensions (VDim1, VDim2, ADim1, ADim2), each taking 2 possible values. Input encoding used 8 dedicated embedding dimensions for stimulus features that were reused between Stim1 and Stim2 presentations. In total, there were six task phases (Rule, Stim1, Dly1, Stim2, Dly2, and Resp). However, there were 10 timepoints in total, as two timepoints were allocated per stimulus and delay period, providing enhanced resolution for studying temporal dynamics.

The compositional structure of CPRO incorporated three types of rule components making up each context. Logical rules included AND, NAND, OR, and NOR operations (4 options). Sensory rules comprised RED, VERTICAL, HIGH-PITCH, and CONSTANT conditions (4 options). Motor rules consisted of LIND, LMID, RIND, and RMID responses (4 options). This design yielded a total task space of 64 unique tasks through all possible combinations of the three rule components ($4 \times 4 \times 4$). All stimulus and context inputs are one-hot encoded.

### A.2.2    TRAINING AND GENERALIZATION SPLITS

We evaluated compositional generalization by training networks on 60/64 contexts (rule combinations) while systematically withholding specific combinations for testing. The test set consisted of

4 contexts where each of the 12 rules occurred exactly once. This test set design required compositional generalization on novel task combinations, enabling assessment of systematic compositional generalization (Hupkes et al., 2020).

**Batch training structure.** Training data incorporated 256 unique stimulus combinations per task context, representing a $16 \times 16$ grid of possible Stim1-Stim2 pairs. We organized training batches such that all samples in a batch belonged to a single task context, following prior precedent in Yang et al. (2019); Driscoll et al. (2024). Training batches were ordered in nested loops across logical, sensory, and motor contexts. This created a specific curriculum structure where logical and sensory contexts remained constant across 4 consecutive motor contexts before transitioning to the next logical-sensory pairing. With a batch size of 128, each task context required exactly 2 consecutive batches per context. Training data comprised of 15,360 examples per epoch (60 tasks $\times$ 256 stimuli).

**Optimization.** Training employed Stochastic Gradient Descent (SGD), using a learning rate of 0.01. Batch sizes were set to 128 for training. Networks were trained for 1000 epochs. We verified that training loss reached comparable levels across all 50 models by the final epoch (Fig. A1A,B). Loss computation was restricted to the response (Resp) timepoint using cross-entropy loss. Network performance was assessed every 100 training epochs using separate test batches of the 4 held-out generalization tasks (Fig. A1C,D).

## A.3 MODEL ARCHITECTURE AND INITIALIZATION

We implemented an RNN architecture with 128 hidden units following the standard recurrent update rule:

$$h_t = \text{ReLU}(\text{LayerNorm}(W_h h_{t-1} + W_x x_t + b_h)) \tag{1}$$

where $W_x \in \mathbb{R}^{128 \times 20}$ transforms 20-dimensional inputs to 128-dimensional hidden representations, and $W_h \in \mathbb{R}^{128 \times 128}$ performs the recurrent processing. Layer normalization was applied before the ReLU nonlinearity. The output stage consisted of a linear transformation followed by softmax:

$$y_t = \text{softmax}(W_y h_t + b_y) \tag{2}$$

where $W_y \in \mathbb{R}^{4 \times 128}$ projects the hidden state to 4-class logits for classification.

Prior studies illustrated that different learning regimes can be induced by altering model initializations (Chizat et al., 2019). To characterize how representational dynamics affect compositional generalization, we systematically varied the initialization scales of recurrent weights while keeping input and output weights at PyTorch's default uniform initialization. We tested 5 logarithmically-spaced initialization scale values in the interval $[0.01, 3.0]$: $\{0.01, 0.04, 0.17, 0.72, 3.0\}$ (Flesch et al., 2022). These scales multiply normally distributed values (with mean 0, i.e., $\mathcal{N}(0, \sigma)$) to create weight distributions with $\sigma \in \{0.0009, 0.0035, 0.015, 0.0636, 0.2652\}$. We trained 10 networks with different random seeds for each scale, yielding 50 networks total. The rich learning condition corresponded to the smallest initialization scale (0.01), while the lazy learning condition used the largest scale (3.0). Recurrent weights were initialized according to Xavier initialization principles (Glorot & Bengio, 2010). Input layer weights maintained standard Gaussian initialization across all conditions to isolate the effects of recurrent weight scaling.

## A.4 REPRESENTATIONAL ANALYSIS METHODS

We used representational similarity analysis (RSA; Kriegeskorte et al. (2008)) to characterize how specific task features contributed to generalization. Neural activation vectors (128 units) were extracted from the recurrent hidden layer at each of the 10 timepoints during task execution. Neural activation vectors were sorted and averaged according to distinct task features. For example, for context representations, all activations corresponding to each unique context were averaged together; unit activations corresponding to specific stimulus conditions were averaged together; unit activations corresponding to trials for a specific response were averaged together.

### A.4.1 FEATURE-SPECIFIC ANALYSES

Neural representations were analyzed across ten distinct timepoints corresponding to different task periods: Rule_only (rule presentation), Stim1_e and Stim1_l (early and late Stim1 processing), Dly1_e and Dly1_l (early and late delay after Stim1), Stim2_e and Stim2_l (early and late Stim2

processing), Dly2_e and Dly2_l (early and late final delay), and Resp (response period). This temporal decomposition provided two timepoints per stimulus and delay period for enhanced resolution of temporal dynamics.

Feature-specific analysis involved decomposing neural activity into distinct functional components through targeted averaging and singular value decomposition (SVD). Stimulus dimensionality was computed by averaging activity across all 64 contexts for each of the 256 stimulus combinations, creating $(256 \times 128)$ matrices. Context dimensionality averaged activity across all 256 stimulus combinations for each of the 64 contexts, yielding $(64 \times 128)$ matrices. Response dimensionality averaged activity across all trials for each of the 4 potential responses, producing $(4 \times 128)$ matrices. Contingency representational dimensionality was derived by averaging activity according to logical rule satisfaction across all trials, creating $(2 \times 128)$ matrices. For closure and integration trials comparison, the activities of the two trial types are separately averaged into respective feature conditions and two sets of feature dimensionalities (one for closure, another for integration cases) are computed. Global dimensionality analysis used the complete $(16{,}384 \times 128)$ matrix representing all trials without feature-specific averaging. Participation ratios were computed from singular value spectra to quantify effective dimensionality on each feature-specific matrix. We computed the participation ratio as a measure of effective dimensionality of hidden representations through singular value decomposition (SVD) of neural activity matrices. The participation ratio was calculated as

$$\text{PR} = \frac{\left(\sum_{i=1}^{s} \sigma_i\right)^2}{\sum_{i=1}^{s} \sigma_i^2}$$

where $\sigma_i$ represents the singular values from SVD decomposition of neural activity matrices. This metric quantifies the effective number of dimensions utilized by neural representations, with higher values indicating more distributed, high-dimensional representations (see Gao et al. (2017) for further details).

### A.4.2 STATISTICAL ANALYSES

Statistical comparisons employed repeated measures ANOVA to assess dimensionality changes across task phases and rich versus lazy learning conditions. We performed 2-way repeated-measures ANOVA (2 Initialization [Rich, Lazy] × 2 timepoints, AnovaRM) for all pairs of timepoints, reporting interaction terms. Post-hoc analyses were carried out using pairwise t-tests between rich versus lazy learning conditions at all 10 timepoints. We corrected for multiple comparisons using false discovery rate (FDR) correction.

To quantify dimensionality-generalization associations, we conducted linear regression analyses using generalization accuracy from all 50 networks (5 initialization scales × 10 seeds) as the dependent variable and dimensionality values as regressors. Regressions were performed on each timepoint separately, with FDR correction across tests. Feature-specific dimensionality analysis incorporated 10 separate regressions (3 features × 10 timepoints) with FDR correction across all tests. This comprehensive statistical framework enabled robust assessment of the relationships between time-dependent representational dimensionality and compositional generalization performance.

### A.5 REPRESENTATIONAL DYNAMICS IN RICH AND LAZY NETWORKS: TEMPORAL EVOLUTION

High-performing and low-performing networks exhibited fundamentally different representational dynamics, with lazy networks increasing global dimensionality over time compared to the dynamic dimensionality patterns in rich networks. Kendall's tau between time and dimensionality (averaged across 10 seeds) was $\tau = 0.911$ (p = 2.98e-05) for lazy networks versus $\tau = -0.156$ (p = 0.6) for rich networks. This non-monotonicity in rich networks was evident as global dimensionality dropped significantly lower closer to the response timepoint compared to lazy networks (FDR-corrected p < 0.001).

Task feature analysis revealed that high-performing rich networks employed a dynamic encoding strategy, exhibiting high stimulus dimensionality during first stimulus encoding (paired t-tests, FDR-corrected p < 0.001) followed by dimensionality compression after encoding both stimuli. Lazy networks maintained their monotonic pattern (Kendall's $\tau = 1.0$, p = 5.5e-07) compared to the more variable rich networks ($\tau = 0.733$, p = 0.002). Context dimensionality in rich networks showed

an inverse relationship with stimulus dimensionality—suppressed during stimulus presentation but rising during delay periods (Spearman correlation of seed-average dimensionalities: $\rho$ = -0.67, p = 0.033)—while lazy networks showed no such pattern ($\rho$ = 0.394, p = 0.26). Despite these fluctuations, rich network dimensionalities remained substantially lower than lazy networks overall (paired t-tests, FDR-corrected p < 0.001). The early rise in response dimensionality in rich networks suggests capacity for early decision-making when feasible (paired t-tests: FDR-corrected p < 0.01 starting from Stim1_l onwards).

## A.6 SUPPLEMENTARY FIGURES

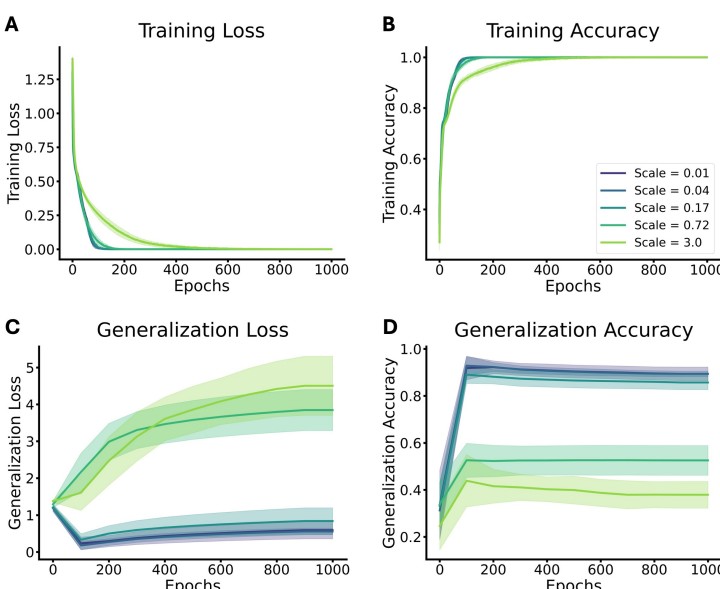

Figure A1: Training and generalization loss and accuracy during training. Evaluation for generalization was performed once every 100 epochs.

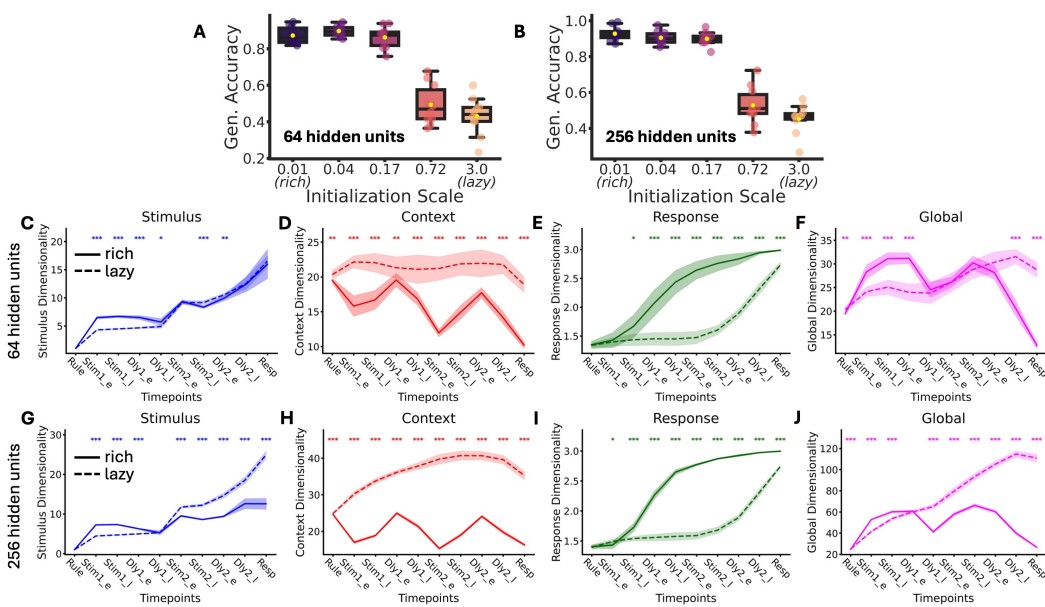

Figure A2: We reproduced the results in the main text using RNNs with 64 (**A**: generalization accuracy, **C-F**: dimensionality traces) and 256 (**B**: generalization accuracy, **G-J**: dimensionality traces) units. Both model variations reproduced patterns similar to the model with 128 units, indicating that these results are robust to model size.

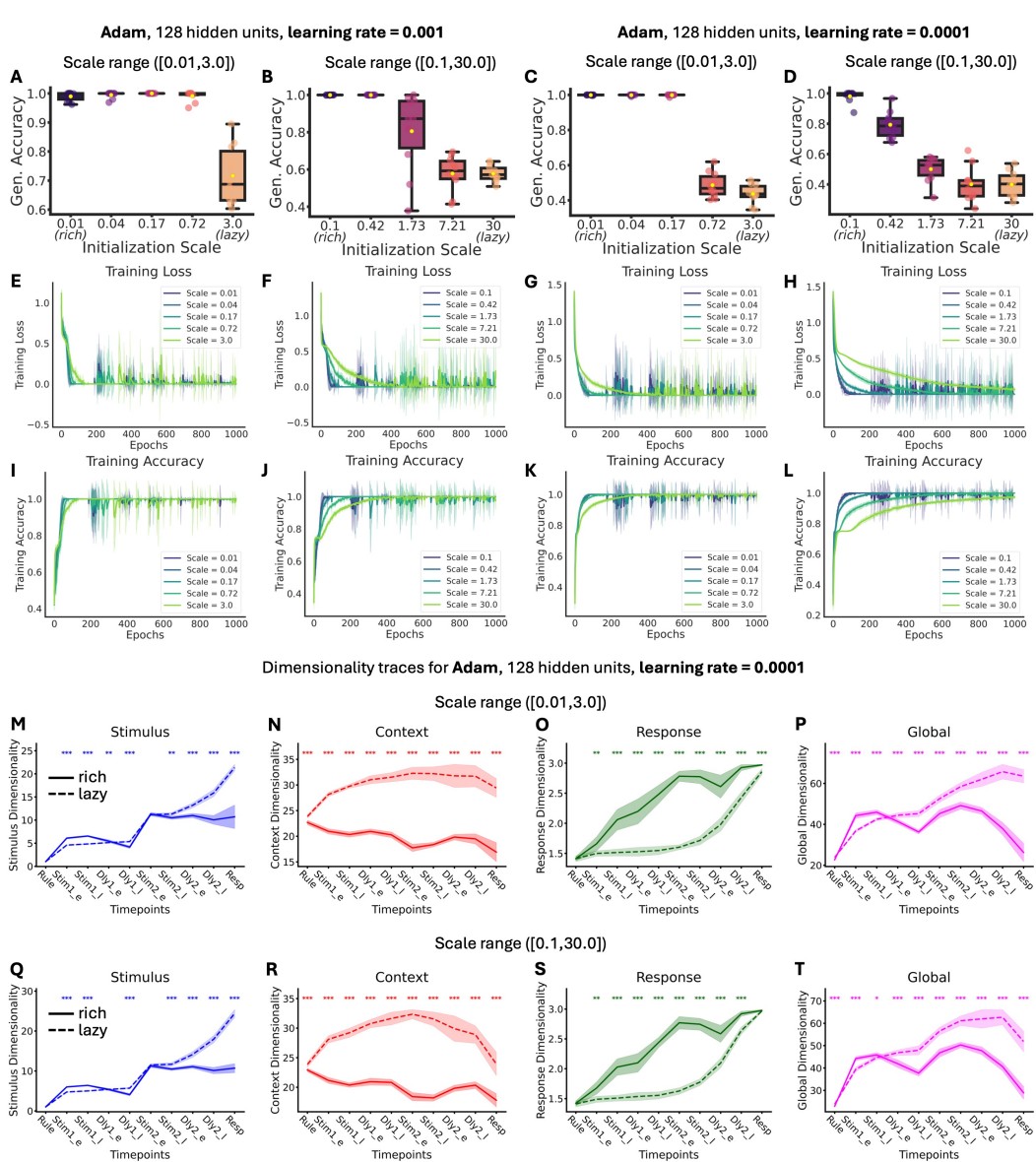

Figure A3: We reproduced the results in the main text using the Adam optimizer (rather than SGD in the main text). We trained networks using Adam optimizer with learning rates of 0.001 (**A, E, F**) and 0.0001 (**C, D, G, H**). **A,C:** Adam achieved near-perfect generalization (≥95%) across the original initialization scale range [0.01, 3.0], eliminating rich-lazy distinctions except at large initializations (e.g., norm of 3.0) (**E, F, G, H**). **B, D:** Prior work suggested that to induce lazy learning with Adam, the initialization scale needs to be magnified (Whitefield & Summerfield, 2025). Thus, as a follow-up, we expanded the initialization range to [0.1, 30.0], where we observed 'lazier' learning with larger initializations.

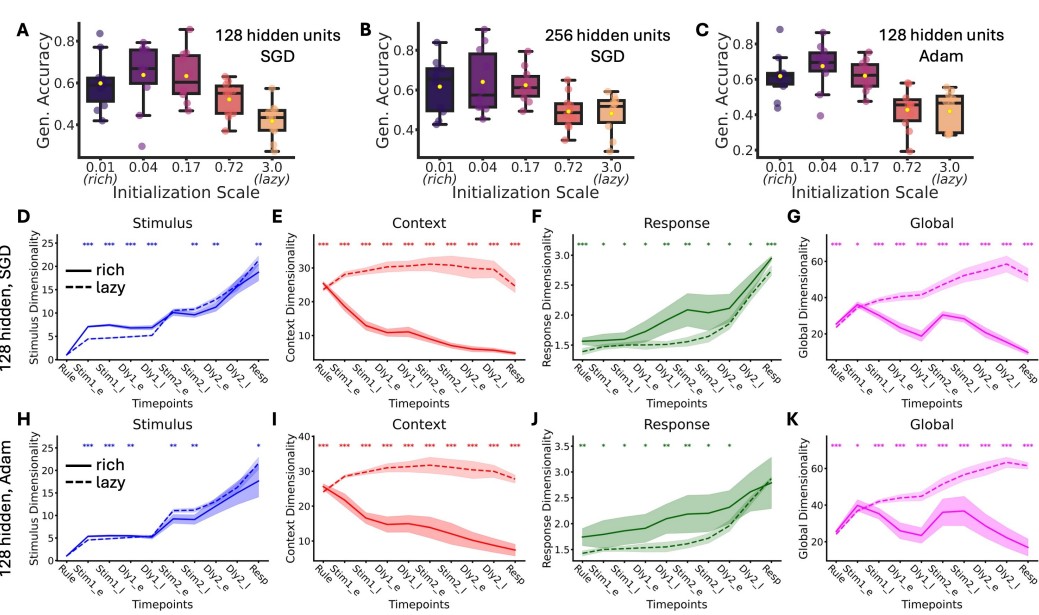

Figure A4: We evaluate model performance with a more challenging task variant, where context information is only available at `Rule` timepoint (i.e., t=0). This increases task difficulty as context information has to be retained over time in the model's working memory. **A)** Richly trained models (initialization scale at 0.01 and 0.04) reached accuracy values close to 65%. (Model parameters were the same as the model used in the main text.) **B)** When the model size was increased to 256 units and number of training epochs increased to 2000, we observed similar results. **C)** Model generalization performance when using the Adam optimizer (learning rate = 0.0001, 1000 epochs). **D-G)** Dimensionality analysis plots for the model with 128 hidden units and SGD. **H-K)** Dimensionality analysis plots for the model with 128 hidden units using the Adam optimizer.

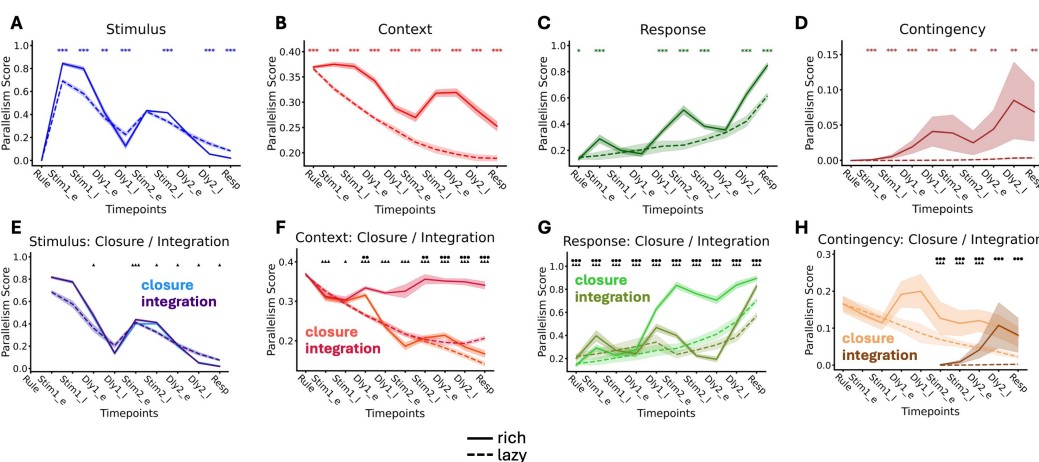

Figure A5: To measure invariance of task representations across contexts, we measured the parallelism score as a measure of representational invariance (or abstraction). Parallelism score measures the alignment of activity vectors across contexts, thereby providing information analogous to cross-condition decoding (the measure was introduced by Bernardi et al. (2020)). Parallelism scores were computed by measuring the cosine similarity between the difference of two vectors, where only a single task feature (rule, stimulus, or response) was varied, while keeping all other task conditions. Specifically, for each variable (e.g., color), we calculated difference vectors between feature values (RED vs BLUE) within the same context, then measured how parallel these vectors are across all context pairs. Higher parallelism indicates more abstract, context-invariant representations. **A,E:** Stimulus abstraction peaks during Stim1 but decreases by Stim2, indicating that stimulus representations become less invariant as integration demands increase. **B,F:** Context information maintains invariant throughout trials, but shows striking differences in closure vs. integration trials – abstraction drops when early decisions are possible, despite being decodable (cf. Figure 4I). Interestingly, this reveals that dimensionality reductions in closure trials reflect loss of invariant representations rather than information loss. **C,G:** Response abstraction increases when decisions can be formed; closure trials achieve high abstraction immediately after Stim1 while integration trials show delayed abstraction. **D,H:** Contingency abstraction mirrors response patterns — closure trials develop abstract True/False representations early and maintain them, while integration trials achieve equivalent abstraction only after Stim2 processing.

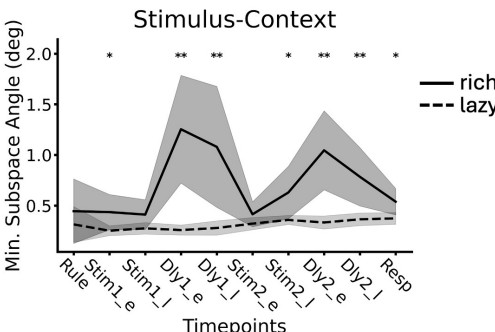

Figure A6: We measured the orthogonality between stimulus and context subspaces, and found that in rich models, stimulus and context subspaces transiently orthogonalized during delay periods. For each feature domain (stimulus and context), we computed subspace bases by using singular value decomposition (SVD) on the averaged neural activity matrices (conditions × neurons). We retained the minimum number of principal components that explained 90% of the variance within each subspace. Principal angles between subspaces were then computed using `scipy.linalg.subspace_angles()`, with the minimum principal angle serving as our index of orthogonality. Principal angles quantify the canonical relationships between subspaces, with the minimum principal angle indicating the most aligned directions between subspaces. We chose to retain the components explaining only 90% of the variance to account for different intrinsic dimensionalities (governed by number of input latent variables) across subspaces. We observed that while the angle is overall close to zero, rich networks demonstrate slight decoupling (increase in the minimum principal angle) during delay periods, whereas lazy models do not.

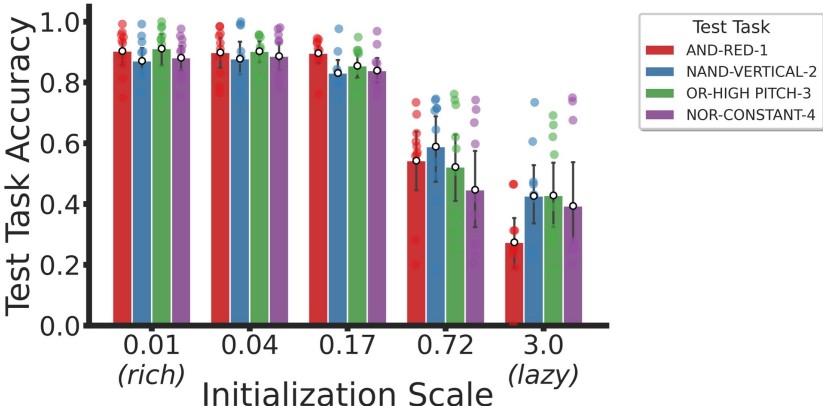

Figure A7: Generalization accuracy by each test context. We observed that rich networks demonstrated similar levels of accuracy when split across the four test contexts. Individual dots represent seeds; error bars show mean ± SD across seeds.

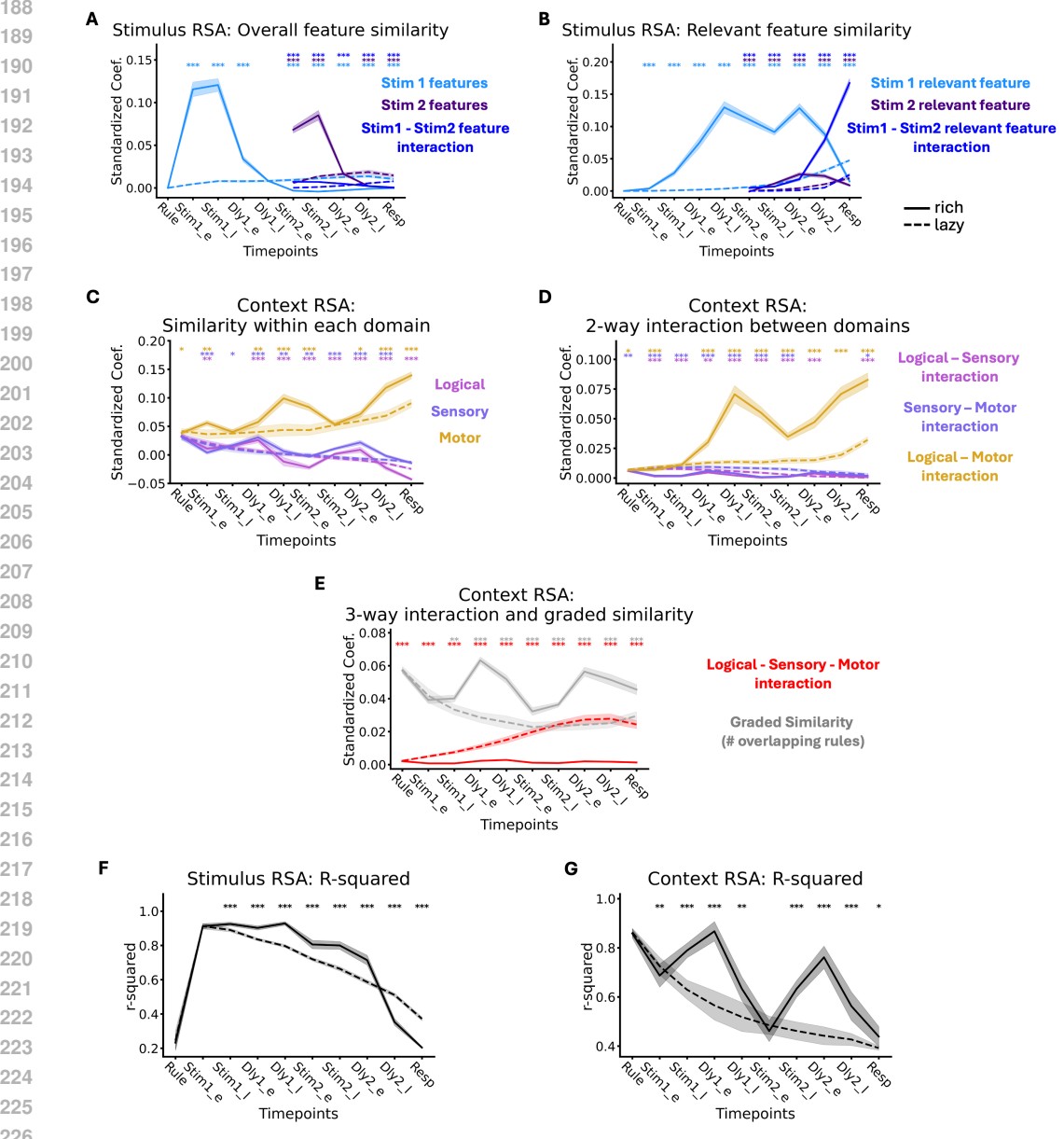

Figure A8: **Stimulus and Context Representational Similarity Analysis (RSA). (A,B,F) Stimulus RSA:** In each context, we computed the similarity matrix among the hidden unit representations of 256 stimulus combinations at each timepoint and tested the following six potential hypotheses together to explain the observed similarity pattern via multiple regression: **A)** Coefficients of *overall stimulus feature similarity* (similarity along the four stimulus dimensions, e.g., whether two stimulus combinations have the same color, orientation etc.) of Stimulus 1, Stimulus 2 and their interaction; **B)** Coefficients of *context-relevant stimulus feature similarity* (similarity along the color dimension if the task context involves evaluation of color dimension of the stimulus) of Stimulus 1, Stimulus 2 and their interaction. **(C-E,G) Context RSA:** In each context, we used the average activity across all stimulus combinations and computed similarity matrix among the 64 context representations at each timepoint and tested the following eight hypotheses to explain the observed similarity pattern via multiple regression: **C)** Coefficients of *similarity within each rule domain*, i.e., whether two contexts have the same logical or sensory or motor rule; **D)** Coefficients of *two-way similarity interactions* among the rule domains; **E)** Coefficients of *three-way similarity interaction* among the rule domains and *graded similarity*, i.e., the number of overlapping rules between two contexts irrespective of the rule domain; **F)** Coefficient of determination (r-squared) of stimulus RSA; **G)** Coefficient of determination of context RSA.

