# OpenReview forum: "Task representational dynamics for compositional generalization"
_ICLR.cc/2026/Conference — ICLR 2026 Conference Withdrawn Submission_

### Official Review · Reviewer_o4aa · 2025-10-14

**Soundness:** 3
**Presentation:** 4
**Contribution:** 2
**Rating:** 6
**Confidence:** 3

**Summary:**

This paper investigates the mechanisms underlying compositional generalization in recurrent neural networks (RNNs) within a temporally structured, context-dependent task (C-PRO). The authors manipulate model initializations to generate networks with a wide range of generalization performance.

The central finding is that successful generalization does not correlate with a single, static representational geometry. Instead, it depends on specific, dynamic modulations of feature-specific representations over the course of a trial. High-performing networks learn to maintain a stable, low-dimensional representation for the task context (rule). During stimulus presentation, the networks expand to a high-dimensional state to integrate stimulus and context information, before compressing the representation into a low-dimensional state for the final response. In contrast, poorly generalizing networks exhibit a monotonic increase in dimensionality, suggesting a less structured, memorization-based strategy.

The primary contributions of this work can be summarized as follows:

A Shift from a Static to a Dynamic Account of Generalization: The work moves beyond prior analyses of static representational structure by demonstrating that the temporal evolution of dimensionality is a key predictor of compositional success. The identified pattern of compression-expansion-compression is a novel finding.

A Proposed Reconciliation of Competing Neuroscientific Theories: The findings offer a computational model that helps reconcile seemingly contradictory results in neuroscience regarding the necessity of both low-dimensional (for abstraction) and high-dimensional (for integration) neural codes, showing they are different phases of a single dynamic process.

A Novel Method for Feature-Specific Representational Analysis: A significant methodological contribution is the decomposition of global network activity into the dynamics of its constituent parts (context, stimulus, response). This technique made it possible to uncover the independent, yet orchestrated, representational trajectories that were obscured by global measures.

Evidence for Emergent, Efficient Cognitive Strategies: The analysis of "closure" and "integration" trials reveals that the learned representational dynamics enable the models to form early decisions when task conditions permit, providing evidence for the emergence of flexible, human-like cognitive strategies rather than a rigid, uniform process.

**Strengths:**

Originality: The paper is reasonably original, viewing the problem of compositional generalization through a different lens (across time and across features), which allows it to find novel insights into the nature of compositional generalization

Quality: The paper's quality is generally good, with clear and concise language, and good contextualization relative to prior work

Clarity: The writing and figures are well presented and designed, as well as grouped together into logical groups of figures

Significance: It is unclear to me (and I admit that I am not very familiar with the literature) whether the authors are trying to make the point that this is how the human brain works, or this is specifically how RNN works, or this is how all AI models work. I think an expanded discussion of the significance of these results for neuroscience/cognitive science/artificial intelligence would significantly strengthen the paper

**Weaknesses:**

1. It is unclear to me (and I admit that I am not very familiar with the literature) whether the authors are trying to make the point that this is how the human brain works, or this is specifically how RNN works, or this is how all AI models work. I think an expanded discussion of the significance of these results for neuroscience/cognitive science/artificial intelligence would significantly strengthen the paper

2. Some of the descriptions of the plots seem to be mischaracterized. For example, line 195: dimensionality tended to monotonically increase across task phases -- this doesn't seem to be true based on the plots presented. I am not disagreeing with the conclusion the authors take from these plots, but I think that the description should be made more carefully

**Questions:**

Suggestions:

1. I think an expanded discussion of the significance of these results for neuroscience/cognitive science/artificial intelligence would significantly strengthen the paper

2. I think the descriptions of the plots in the text for figure 2 should be revised to be more accurate, without changing the conclusions.

---

> ### Author Response · Authors · 2025-11-25
> **Response to Reviewer o4aa**
>
> > **4A (weakness): “It is unclear to me (and I admit that I am not very familiar with the literature) whether the authors are trying to make the point that this is how the human brain works, or this is specifically how RNN works, or this is how all AI models work. I think an expanded discussion of the significance of these results for neuroscience/cognitive science/artificial intelligence would significantly strengthen the paper.”**
>
> **Reply 4A.** We thank the Reviewer for urging us to clarify the scope of this work. We have revised the title to **‘Task representational dynamics for compositional generalization during context-dependent cognitive tasks’** to clarify the scope. We intend this study to be more applicable to cognitive science and neuroscience, focusing on how representation dynamics in cognitive tasks that require temporal evolution relate to generalizable performance. We have also revised a sentence in the abstract to indicate that we are interested in “cognitive” tasks: “Here we investigate how representational dynamics shape compositional generalization in recurrent neural network (RNN) models during cognitive tasks with evolving temporal structure.”
>
> In addition, we have significantly revised the Discussion’s Related Works section (Section 4.1). We specifically relate works from computational neuroscience modeling work (**“Representational dynamics in models for computational neuroscience”**, **“Rich and lazy learning regimes”**) and empirical neuroscience work **“Relation to empirical neuroscience and cognitive science”**.
>
> > **4B (weakness):“Some of the descriptions of the plots seem to be mischaracterized. For example, line 195: dimensionality tended to monotonically increase across task phases -- this doesn't seem to be true based on the plots presented. I am not disagreeing with the conclusion the authors take from these plots, but I think that the description should be made more carefully.”**
>
> **Reply 4B.** Thanks to the Reviewer for pointing this out. We have removed the term “monotonically” from this phrase.
>
> > **4C (suggestion): “I think an expanded discussion of the significance of these results for neuroscience/cognitive science/artificial intelligence would significantly strengthen the paper.”**
>
> **Reply 4C.** We thank the reviewer for the suggestion. We hope we addressed it in response **4A** above. Note that the revisions in the updated PDF are noted in blue text.
>
> > **4D (suggestion): “I think the descriptions of the plots in the text for figure 2 should be revised to be more accurate, without changing the conclusions.”**
>
> **Reply 4D**. We thank the reviewer for the suggestion. We hope we addressed it in response **4B** above.
>
> **References:**
> * Chizat, L., Oyallon, E., & Bach, F. (2019). On lazy training in differentiable programming. Advances in neural information processing systems, 32.
> * Flesch, T., Juechems, K., Dumbalska, T., Saxe, A., & Summerfield, C. (2022). Orthogonal representations for robust context-dependent task performance in brains and neural networks. Neuron, 110(7), 1258-1270.
> * Khona, M., Chandra, S., Ma, J. J., & Fiete, I. R. (2023). Winning the lottery with neural connectivity constraints: Faster learning across cognitive tasks with spatially constrained sparse rnns. Neural Computation, 35(11), 1850-1869.

---

### Official Review · Reviewer_xC1Z · 2025-10-28

**Soundness:** 2
**Presentation:** 2
**Contribution:** 2
**Rating:** 4
**Confidence:** 4

**Summary:**

This paper studies the representations developed in RNNs to solve a compositional generalization task. Specifically, the authors compare the performance and the temporal dynamics of latent representations in RNNs initialized with rich or lazy learning regime (different initialization norms). The results yield insights into the temporal changes in the representation dimensionality for rule/context, stimulus, and response information, and how the dimensionality changes link to compositional generalization accuracy.

**Strengths:**

- The temporal changes in representation dimensionality provide a nice unification of prior mixed findings in neuroscience.
- This work provides quite a comprehensive analysis of the model's learned representational strategy in a compositional generalization task.

**Weaknesses:**

- I find it a bit unclear what the target scope/audience for the work is. At the outset, this work seems to target the general issue of compositional generalization in RNNs. But later on it seems that the results speak more directly to reconciling prior conflicting findings in neuroscience. I think an effort to clarify the scope in the title/abstract could help resolve this potential confusion.
- Related to the above, as acknowledged, the results are limited to a particular task and model setup. This seems less of an issue if the goal is to provide a resolution to prior conflicting findings in neuroscience, but more of a concern if the goal is to speak to compositional generalization abilities in RNNs at large.
- It's a bit unclear what representational dimensionality mechanistically capture -- sometimes low dimensionality is attributed with abstract representation, other times attributed with reduced information. I think some causal exploration of these hypotheses, or additional clarity on whether rep. dim. reflects available information vs. information used downstream, would greatly enhance the paper.
- The clarity in writing can be improved. Specifically, more pointers to the the architectural details can be good. The key metric studied, representation dimensionality, also seems to be missing a definition in the main text.

**Questions:**

- Have the authors explored task/model variations, and whether these results generalize across different settings? For example, what may happen if the rule signal is given after the stimulus presentation, or if the hidden layer size is increased or decreased?
- Can the same regression analysis and closure/integration trial comparison be done on the lazy model? This seems like an interesting opportunity to reveal insights into the model's failure mode.

---

> ### Author Response · Authors · 2025-11-25
> **Response to Reviewer xC1Z (1/N)**
>
> > **3A (weakness): “I find it a bit unclear what the target scope/audience for the work is. At the outset, this work seems to target the general issue of compositional generalization in RNNs. But later on it seems that the results speak more directly to reconciling prior conflicting findings in neuroscience. I think an effort to clarify the scope in the title/abstract could help resolve this potential confusion.”**
>
> **Reply 3A.** We thank the Reviewer for urging us to clarify the scope of this work. We have revised the title to **‘Task representational dynamics for compositional generalization during context-dependent cognitive tasks’**. We intended this manuscript to be more suitable for cognitive science and neuroscience. We have also revised a sentence in the abstract to indicate that we are interested in “cognitive” tasks: “Here we investigate how representational dynamics shape compositional generalization in recurrent neural network (RNN) models during **cognitive** tasks with evolving temporal structure.”
>
> In addition, we have significantly revised the Discussion’s Related Works section (Section 4.1). We specifically relate works from computational neuroscience modeling work (“**Representational dynamics in models for computational neuroscience**”, “**Rich and lazy learning regimes**”) and empirical neuroscience work “**Relation to empirical neuroscience and cognitive science**”.
>
> > **3B (weakness): “Related to the above, as acknowledged, the results are limited to a particular task and model setup. This seems less of an issue if the goal is to provide a resolution to prior conflicting findings in neuroscience, but more of a concern if the goal is to speak to compositional generalization abilities in RNNs at large.”**
>
> **Reply 3B.** We hope that the response above helps to address this concern. However, to address the concerns of this and other reviewers, we have also performed numerous additional analyses aimed to evaluate our hypotheses under additional model/optimization/task variations. These major additions are reported in **Response 0B** in response to all Reviewers.
>
> > **3C (weakness): “It's a bit unclear what representational dimensionality mechanistically capture -- sometimes low dimensionality is attributed with abstract representation, other times attributed with reduced information. I think some causal exploration of these hypotheses, or additional clarity on whether rep. dim. reflects available information vs. information used downstream, would greatly enhance the paper.”**
>
> **Reply 3C.** This was an important comment that we have worked to clarify with numerous additional analyses. We recognize that dimensionality alone isn’t enough to characterize what information is available, and whether that information is used or invariant (or abstracted) from that specific context. Therefore we complemented the dimensionality analyses with several direct task decoding analyses. **We have included: 1) Temporal decoding analyses for each task feature (Appendix Fig. A5, A6) where we decode task features at each time point, and 2) Cross-context parallelism score (similar to cross-condition decoding; Appendix Fig. A8) to assess representational invariance across contexts.** To summarize one interesting finding from these new analyses (though please refer to the referenced Figures for the full set of results): One decoding result indicates that high stimulus dimensionality during stimulus period better encodes stimulus information (**Fig. A5A, A5E**), whereas lower context dimensionality (relative to lazy networks) indicates better context decoding (**Fig. A5B, A5F**). This indicates that high dimensionality does not necessitate strong information decoding for that task feature. Please see Fig. A5, A6, A7, and A8 for the full set of decoding/cross-condition analyses, with results detailed in the caption. By directly decoding task features from the neural representations (not only measuring dimensionality) we hope these provide more clear evidence regarding how dimensionality reflects usable/available information within the neural representations. In the next revision, we plan to integrate these results with the main text to clarify the interpretation of dimensionality where applicable. Thank you for this suggestion, as we believe they significantly enhance the interpretability of the other key results.
>
> > **3D (weakness): “The clarity in writing can be improved. Specifically, more pointers to the architectural details can be good. The key metric studied, representation dimensionality, also seems to be missing a definition in the main text.”**
>
> **Reply 3D.** We thank the reviewer for this suggestion. We have revised and simplified Section 2.4 to provide a clearer description of both the model and how representational dimensionality was computed. We have also included a revision of Section 2.1 to include a more thorough description of the Task modeling set up.

---

> ### Author Response · Authors · 2025-11-25
> **Response to Reviewer xC1Z (N/N)**
>
> > **3E (question): “Have the authors explored task/model variations, and whether these results generalize across different settings? For example, what may happen if the rule signal is given after the stimulus presentation, or if the hidden layer size is increased or decreased?”**
>
> **Reply 3E.** Thanks to the Reviewer for this suggestion. We performed additional results that evaluate on both model and task variations. For model variations, we manipulated the number of hidden units, and reproduced the main analyses (64, 128 (original), and 256; **Fig. A2)**. We also included a variant where we optimized the model using Adam, rather than vanilla SGD **(Fig. A3)**. For these model variations, we overall found that generalization accuracy and dimensionality dynamics to be consistent and qualitatively similar to our original results. For task variation, we tested a variant where context was only presented at the initial timepoint, i.e., t=0, rather than being present throughout (consistent with prior modeling efforts in neuroscience; Yang & Wang, 2020). Overall, this task variant turned out to be more challenging for the models to generalize to (~65% accuracy; **Fig. A4**), likely because it required the model to maintain context information in the model’s working memory.
>
> While it would be interesting to explore task variations in which the context comes after the stimuli, as the Reviewer suggested, this would increase the difficulty of the task. Moreover, it would alter the cognitive capability being assessed, making it resemble a working memory selection task (e.g., Panichello & Buschman, 2019). Under this configuration, we would no longer be able to characterize closure and integration conditions, as all stimulus features would need to be maintained in the model’s recurrent state – features we consider central to the current paradigm. That said, we agree that this variant represents an intriguing direction for future work.
>
> > **3F (question): “Can the same regression analysis and closure/integration trial comparison be done on the lazy model? This seems like an interesting opportunity to reveal insights into the model's failure mode.”**
>
> **Reply 3F.** Thank you for this important suggestion. When we performed the regression analysis, linking fluctuations in dimensionality with generalization performance at each time point, this included the full distribution of models, including models from all initializations. This distribution allowed for enough variation to identify whether dimensionality was actually predictive of generalization. Indeed, the Reviewer suggests an interesting analysis – to show these plots for integration and closure trials separately. Though we have not included this specific analysis, we instead performed new analyses that illustrate the decodability traces for closure/integration conditions in **Fig. A7**. We hope that these traces provide clarity regarding what information content actually is present in each model across closure/integration trials.
>
> **References:**
> * Panichello & Buschman, 2019: Panichello, M. F., & Buschman, T. J. (2021). Shared mechanisms underlie the control of working memory and attention. Nature, 592(7855), 601-605.
> * Yang & Wang, 2020: Yang, G. R., & Wang, X. J. (2020). Artificial neural networks for neuroscientists: a primer. Neuron, 107(6), 1048-1070.

---

> > ### Comment · Reviewer_xC1Z · 2025-11-27
> >
> > I thank the authors for the detailed response. I think these additional changes and experiments were helpful additions to improve the technical clarity and soundness of the paper. I have thus increased my ratings in those areas. However, I still have some overall reservations regarding its scope and generality. Specifically:
> > - I still find that the paper could be much more explicit in stating whether/how these studies serve as models of human behavior or the brain or enlist its particular insights towards interpreting human/animal data, rather than addressing the issue of compositional generalization in RNNs or NNs at large.
> > - I think the new results should be significantly integrated into the main text to complement the dimensionality analyses.
> > - I have some quibbles about the notion of studying "cognitive" tasks. As acknowledged, the C-PRO task is operationalized in a way that abstracts away much of the realistic and multimodal components in the human task. This makes learning from this particular setting limited in its implications to cognitive processes or operations.
> >
> > I thus intend to keep my overall score.

---

> > > ### Author Response · Authors · 2025-12-03
> > > **Response to Official Comment by Reviewer xC1Z (1/N)**
> > >
> > > **We thank the Reviewer for replying to our initial rebuttal**
> > >
> > > > **3G (comment): ”I think these additional changes and experiments were helpful additions to improve the technical clarity and soundness of the paper. I have thus increased my ratings in those areas.”**
> > >
> > > **Reply 3G.** We thank the Reviewer for acknowledging the value of the additional changes and results from new analyses and experiments, particularly in regards to their technical clarity and soundness of the results.
> > >
> > > > **3H (comment): ”I still find that the paper could be much more explicit in stating whether/how these studies serve as models of human behavior or the brain or enlist its particular insights towards interpreting human/animal data, rather than addressing the issue of compositional generalization in RNNs or NNs at large.”**
> > >
> > > **Reply 3H.** In our second revision, we have clarified the scope in the abstract by adding these sentences: “Here we investigate how representational dynamics shape compositional generalization in recurrent neural network (RNN) models during cognitive tasks with evolving temporal structure, **providing mechanistic insights into neural computation during flexible reasoning.**” and “... **—providing testable principles for how neural systems enable flexible reasoning.**” towards the end.
> > >
> > > Furthermore, in revising the manuscript, we have made a number of edits and deletions to the text that referenced compositionality in NNs at large that the Reviewer may have missed (as the removal of text is often hard to spot). Finally, while our last revision significantly revised the Related Works section within the Discussion to recontextualize our work in within the context of neuroscience and cognitive science, we have made additional revisions to the section related our work to the broader compositionality in ML literature:
> > >
> > >
> > > “Our work differs from traditional compositionality studies in ML by focusing our efforts on temporally-structured compositional paradigms established in the human literature to address compositionality in cognitive science. While our experiments focus on a single task and architecture (RNNs), future work should assess generalizability across diverse cognitive demands, network architectures, and human data.”
> > >
> > > > **3I (comment): ”I think the new results should be significantly integrated into the main text to complement the dimensionality analyses.”**
> > >
> > > **Reply 3I.** We thank the Reviewer for the suggestion. In the second revision of the manuscript, we made significant changes to the **sections 3.1, 3.2.2, 3.3.1 and 3.3.2** to integrate the results from decoding, RSA, parallelism and subspace angle analyses, with references in the Appendix to experiments on model size, optimizer and task variations. The main text now references all new Appendix Figures and the analyses therein. Revised text for the second revision can be seen in purple.

---

> > > ### Author Response · Authors · 2025-12-03
> > > **Response to Official Comment by Reviewer xC1Z (N/N)**
> > >
> > > > **3J (comment): ”I have some quibbles about the notion of studying "cognitive" tasks. As acknowledged, the C-PRO task is operationalized in a way that abstracts away much of the realistic and multimodal components in the human task. This makes learning from this particular setting limited in its implications to cognitive processes or operations.”**
> > >
> > >
> > > **Reply 3J.** We recognize that while our implementation uses symbolic encodings rather than naturalistic multimodal stimuli, **the core cognitive operations are preserved**: selective attention to specific features based on context, working memory integration across delay periods, context-dependent rule application, and compositional recombination of learned rule components – the fundamental operations C-PRO was designed to probe. In our simplified task design, operations that are not explicitly related to compositional reasoning, such as sensory perception, are abstracted away. *Importantly, this approach to experimental design for computational models in neuroscience – reducing complex stimuli, contexts, and rules to simplified input representations – follows standard practice in computational modeling (Yang et al., 2019; Driscoll et al., 2024; Johnston & Fusi 2024). This allows modelers to isolate cognitive mechanisms while maintaining experimental control.
> > > We have now included a new Limitations section in the Discussion (Section 4.2) to more explicitly lay out the limitations of the current study:
> > > > “Our study highlights the critical role of structured temporal representations in enabling robust compositional generalization in RNNs. Nonetheless, several limitations remain in the present work. First, while our implementation of the C-PRO task preserves its core cognitive operations -- compositional reasoning, working memory integration, and contingency formation -- we used symbolic input encodings (e.g., multi-hot encodings of binary features) rather than naturalistic stimuli used in the original human task design. While this limits direct comparisons to the naturalistic sensory processes involved in human cognition, this allowed us to isolate the cognitive processes of interest (compositional reasoning). Nevertheless, it will be interesting for future work to explore the temporal evolution of representations in compositional tasks that use naturalistic stimuli. Second, we studied the contribution of representational dynamics to compositional generalization by manipulating RNN weight initialization. While prior work has demonstrated that weight initialization can strongly influence generalization (Chizat al., 2019; Farrell et al., 2023), it is possible that there are other model manipulations -- as well as model architectures and optimization protocols -- that induce different relationships between representational dynamics and generalization that future work can explore. Finally, we limited our investigation to the C-PRO task. While the C-PRO task has a highly stereotyped task configuration that is commonly-used within neuroscience and cognitive science (i.e., context -> stimulus -> response), it will be important to explore compositional task designs in future work.
> > >
> > > **References:**
> > > * Driscoll, L. N., Shenoy, K., & Sussillo, D. (2024). Flexible multitask computation in recurrent networks utilizes shared dynamical motifs. Nature Neuroscience, 27(7), 1349-1363.
> > > * Johnston, W. J., & Fusi, S. (2023). Abstract representations emerge naturally in neural networks trained to perform multiple tasks. Nature Communications, 14(1), 1040.
> > > * Yang, G. R., Joglekar, M. R., Song, H. F., Newsome, W. T., & Wang, X. J. (2019). Task representations in neural networks trained to perform many cognitive tasks. Nature neuroscience, 22(2), 297-306.

---

### Official Review · Reviewer_zxke · 2025-11-01

**Soundness:** 3
**Presentation:** 4
**Contribution:** 3
**Rating:** 6
**Confidence:** 4

**Summary:**

The authors trained a range of RNNs on a compositional generalization benchmark, C-PRO. They find that rich RNNs generalized substantially better than lazy RNNs. They then analyzed the dimensionality of the stimulus, context, and response representations identifying specific transitions between high and low-dimensional representations over time. In particular, their analysis implies that at different times the rich networks expand and compress dimensionality. Further, they found substantially different dynamics between trials that required integrating both stimuli and trials that did not.

**Strengths:**

Overall, I liked this paper. I thought it was well-written and covers an important topic: how does feature learning in RNNs influence compositional generalization and what are the representational dynamics of a network that can generalize compositionally? The authors provide a careful and detailed analysis of these representational dynamics, focusing on analyzing the participation ratio. In particular, I think the insight that different phases of the task require either high or low dimensionality is really interesting and provides further nuance to the common wisdom that rich learning induces low dimensionality and that low dimensionality is good. Overall, this paper could be a valuable contribution to ICLR and the general literature on compositional generalization, in particular in a neuroscience context where RNNs are an important architecture that is generally understudied in the context of compositional generalization.

**Weaknesses:**

Primarily, I think the paper would benefit substantially from additional analyses that go beyond just analyzing the participation ratio and further investigate what those different dimensions are representing. For example, representations could be analyzed by looking at representational similarity matrices in more depth, by looking at whether information is invariantly encoded across conditions (e.g. by analyzing cross-condition generalization performance, see e.g. Bernardi et al., 2020), or by plotting representations using dimensionality reduction methods. Below I'm highlighting a few questions the current analysis leaves open that I think such an analysis could answer:

- What information do the high-dimensional stimulus encodings represent in contrast to the low-dimensional stimulus encodings?
- How are the different contexts encoded in a lower-dimensional space?
- How do the encoding of stimulus and context interact across the different timesteps? Are they orthogonal to each other or are they entangled with each other?
- In closure conditions is the correct response already encoded invariantly to e.g. context before the second stimulus is perceived?

In my opinion, this would substantially improve our understanding of how the rich RNN solves this task. I want to emphasize that the paper has several interesting takeaways and provides a sound analysis, so I think it could be a valuable contribution in its current form. However, I think it also leaves important questions about the representational dynamics in the RNNs unanswered, which limits its impact. I would therefore currently consider the paper a "marginal accept".

**Questions:**

- Can your current analyses address the questions I highlighted in "Weaknesses"?
- Why do you keep the task context active throughout? Do you anticipate that that strongly changes network behavior?
- In the rich networks, is there any structure to the errors the network continues making? Are the errors made randomly or are there certain held-out data these networks consistently don’t generalize on? What do you think causes the sub-100% generalization?
- Does your analysis make predictions that could be tested in the fMRI recordings or is the temporal resolution of fMRI too low?

**Minor suggestions**
- Minor point: You're saying that "lazy" learning memorizes (l. 38). I see what you're saying but I would avoid the term "memorization". There are many cases where lazy models generalize, including to out-of-distribution data [1] and to compositional generalization data [2]. Similarly, while Lippl & Stachenfeld (which you cite earlier in that sentence) show that lazy models can't achieve certain generalizations, they also show that lazy models are capable of other kinds of compositional generalization.
- You may find [3] a relevant paper as it studies RNNs solving a task involving compositional generalization (transitive inference).
- You may find [4] a relevant paper, as it studies the impact of rich-regime learning on compositional generalization.
- I think there’s a typo here: “ while other studies have suggested that abstraction – which often requires learning compressed, lower dimensional representations – are more amenable to generalizable task performance” — maybe “abstraction” -> “abstract representations”?
- l. 265: Do you mean “at each timepoint”?

1. https://proceedings.neurips.cc/paper/2021/hash/691dcb1d65f31967a874d18383b9da75-Abstract.html
2. https://www.jmlr.org/papers/v25/24-0220.html
3. https://journals.plos.org/ploscompbiol/article?id=10.1371/journal.pcbi.1011954
4. https://arxiv.org/abs/2503.09781

---

> ### Author Response · Authors · 2025-11-25
> **Response to Reviewer zxke (1/N)**
>
> > **2A (weakness): “Primarily, I think the paper would benefit substantially from additional analyses that go beyond just analyzing the participation ratio and further investigate what those different dimensions are representing. For example, representations could be analyzed by looking at representational similarity matrices in more depth, by looking at whether information is invariantly encoded across conditions (e.g. by analyzing cross-condition generalization performance, see e.g. Bernardi et al., 2020), or by plotting representations using dimensionality reduction methods. Below I'm highlighting a few questions the current analysis leaves open that I think such an analysis could answer:”**
>
> **Reply 2A**. We thank the Reviewer for making these insightful suggestions. Following their suggestions, we performed a number of new analyses to further investigate what information the neural representations contain. **1) Temporal decoding analyses for each task feature (Appendix Fig. A5, A6), 2) Cross-context parallelism score (similar to cross-condition decoding; Appendix Fig. A8), and 3) Computing the principal angle between task context and stimulus subspaces Appendix Fig. A9).**
>
> We are also in the process of performing an RSA-based analysis, as suggested by the Reviewer. Overall the results are consistent with what we have found with decoding analyses. We intend to report the RSA results in the next round of updates/revisions.
>
> Overall, our results corroborate and complement the previous dimensionality analyses, adding depth to our understanding of exactly what information is decodable across task phases, and whether the representations are themselves generalizable (cross-condition invariance). We expand on the relevant results in responses below.
>
>
> > **2B (weakness): “What information do the high-dimensional stimulus encodings represent in contrast to the low-dimensional stimulus encodings?”**
>
> **Reply 2B.** Results from our new decoding results suggest that in high-performing (rich) networks, high stimulus dimensionality during stimulus presentation indicates better encoding of all stimulus dimensions (hence high decoding, **Fig. A5A, A5E**). Interestingly, we have included an additional analysis that illustrates that the drop in dimensionality during delay period signifies selection of context-relevant stimulus features (e.g., gating of relevant features only) (**Fig. A6**). For brevity here, we refer the Reviewer to additional details regarding these results in their respective figure captions.
>
>
> > **2C (weakness): “How are the different contexts encoded in a lower-dimensional space?”**
>
> **Reply 2C.** While we find that higher stimulus dimensionality typically is associated with decodability, we observe distinct patterns with context decoding (see **Fig. A5B, A5F**). In particular, we find that low context dimensionality is accompanied by high context decodability, when comparing rich versus lazy models (i.e., contrast rich and lazy networks in **Fig. A5B  and Fig. A5F**).
>
> Cross-condition analyses (where we used the parallelism score to quantify representational invariance) illustrate that low-dimensionality of context representations is accompanied by greater cross-condition invariance/abstraction in rich models **(Fig. A8B, A8F)**. In short, parallelism score measures the alignment of activity vectors across contexts, thereby providing information equivalent to cross-condition decoding (the measure was introduced by Bernardi et al., 2020). Naturally, in lazy models, we do not observe high cross-condition invariance, as measured by low parallelism.
>
>
> > **2D (weakness): “How do the encoding of stimulus and context interact across the different timesteps? Are they orthogonal to each other or are they entangled with each other?”**
>
> Reply 2D. We thank the Reviewer for an interesting analysis suggestion. To investigate whether stimulus/context are aligned/orthogonal across time steps, we computed the minimum principal angle (indicating the most aligned directions between subspaces) between the stimulus and context principal subspaces (Fig. A9). Interestingly, we found that rich networks exhibit slight decoupling (angle/orthogonality increases) between the context and stimulus vectors during the delay periods, while this behavior is absent in lazy networks. We are also in the process of running RSA to better characterize stimulus-context interaction patterns. This analysis provides a dynamic view into how context and stimulus information evolve (and interact) over time.

---

> ### Author Response · Authors · 2025-11-25
> **Response to Reviewer zxke (2/N)**
>
> > **2E (weakness): “In closure conditions is the correct response already encoded invariantly to e.g. context before the second stimulus is perceived?”**
>
> **Reply 2E.** To check for invariance of the intermediate response information in closure and integration conditions, we computed parallelism scores for the two measures across contexts (**Fig. A8 G-H, detailed description of the method in figure caption**). To do so, we estimated the difference between vectors in the true and false contingency representations in one task context, and evaluated whether they were parallel to the same difference vector in a different context. We noticed that the parallelism score was higher in closure conditions after Delay1 for both intermediate True/False representations, as well as response representations, suggesting invariant encodings in closure conditions (as the Reviewer alludes to).
>
> > **In my opinion, this would substantially improve our understanding of how the rich RNN solves this task. I want to emphasize that the paper has several interesting takeaways and provides a sound analysis, so I think it could be a valuable contribution in its current form. However, I think it also leaves important questions about the representational dynamics in the RNNs unanswered, which limits its impact. I would therefore currently consider the paper a "marginal accept".**
>
> We thank the Reviewer for their overall positive assessment of our manuscript. We hope that the additional analyses and extensive revisions have addressed their concerns!
>
> > **2F (question): “Can your current analyses address the questions I highlighted in "Weaknesses"?”**
>
> **Reply 2F.** We are grateful to the Reviewer for their constructive and thoughtful Review. In addressing each comment – particularly the inclusion of temporal decoding analyses and parallelism score analyses – we believe we have significantly improved the manuscript and clarified key aspects of what the representational dimensionality actually measures. Because we believe the temporal decoding analyses are so intuitive (and complementary with dimensionality), we intend to include those results (currently in Appendix Fig. A5) into the main text for the next set of revisions.
>
> > **2G (question): “Why do you keep the task context active throughout? Do you anticipate that that strongly changes network behavior?”**
>
> **Reply 2G.** The primary reason we decided to keep the task context active throughout was due to prior precedent in the cognitive modeling literature (Yang et al., 2019; Yang & Wang 2020). Nevertheless, to address the Reviewer’s concern, we performed an additional experiment where we constrained the context input to the network to only the initial timepoint (i.e., t=0). This variant of the task turned out to be more difficult to optimize and generalize (generalization peaked at ~65%, even for rich models). This is because the context needed to be retained in the model’s ‘working memory’. However, we have now reported these results in the Appendix, as well as providing representational decoding/dimensionality plots in new figures (**Fig. A4**).
>
> > **2H (question): “In the rich networks, is there any structure to the errors the network continues making? Are the errors made randomly or are there certain held-out data these networks consistently don’t generalize on? What do you think causes the sub-100% generalization?”**
>
> **Reply 2H.** Model generalization achieves 90% accuracy. Note that this performance implies systematic compositional generalization, which requires the model to perform the task on a new combination of rules to form a completely novel context. However, we computed the generalization performance across these new contexts to see whether models failed to generalize in a systematic way (e.g., worse performance on a specific set of rules) **(Fig. A10). Overall, we found no significant differences in performance when evaluating performance on each rule separately.** In general, we believe this form of generalization (i.e., systematic out-of-distribution generalization) remains a challenging problem for RNNs, particularly for tasks that evolve across time.
>
> > **2I (question): “Does your analysis make predictions that could be tested in the fMRI recordings or is the temporal resolution of fMRI too low?”**
>
> Reply 2I. Indeed, our results make tractable predictions for how representational dynamics would unfold across high vs. low performers (e.g., rich vs. lazy). While the temporal resolution of fMRI is low, it may still be able to test these empirically using a task design that has appropriately spaced out temporal windows for each task phase. (These hypotheses could also be readily tested with EEG, albeit with lower spatial resolution.) We appreciate the comment, as we believe it will be interesting to explore these predictions in a future study.

---

> ### Author Response · Authors · 2025-11-25
> **Response to Reviewer zxke (N/N)**
>
> > **2J (suggestion): “Minor point: You're saying that "lazy" learning memorizes (l. 38). I see what you're saying but I would avoid the term "memorization". There are many cases where lazy models generalize, including to out-of-distribution data [1] and to compositional generalization data [2]. Similarly, while Lippl & Stachenfeld (which you cite earlier in that sentence) show that lazy models can't achieve certain generalizations, they also show that lazy models are capable of other kinds of compositional generalization.”**
>
> **Reply 2J.** Thank you for the suggestion. We have removed the term ‘memorization’. The revised sentence (Section 1):
>
> >“Other theoretical work has identified distinct learning regimes, with "rich" learning leading to structured representations that likely enhance compositional generalization (Lippl & Stachenfeld, 2024), while "lazy" models learn input-output mappings by projecting input features to a random, high-dimensional space, similar to reservoir computing.”
>
> We also added the two references the Reviewer suggested in the Discussion (Section 4.1):
>
> > “Studies have addressed the issue of improving generalization even in the lazy learning regime (Canatar et al., 2021; Abbe et al., 2024).”
>
> > **2K (suggestion): “You may find [3] a relevant paper as it studies RNNs solving a task involving compositional generalization (transitive inference). You may find [4] a relevant paper, as it studies the impact of rich-regime learning on compositional generalization.”**
>
> **Reply 2K.** Thank you for the suggested references, which we had inadvertently omitted. We have included them in the revised Section 4.1.
>
> > **2L and 2M (suggestions): “I think there’s a typo here: “ while other studies have suggested that abstraction – which often requires learning compressed, lower dimensional representations – are more amenable to generalizable task performance” — maybe “abstraction” -> “abstract representations”? l. 265: Do you mean “at each timepoint”?”**
>
> **Reply 2L.** Thank you – we made the suggested change from **abstraction** to **abstract representations** in the main text.
>
> **Reply 2M.** Yes – we meant “at **each** timepoint” and have made this change in the text.
>
>
> **References:**
> * Bernardi et al., 2020: Bernardi, S., Benna, M. K., Rigotti, M., Munuera, J., Fusi, S., & Salzman, C. D. (2020). The geometry of abstraction in the hippocampus and prefrontal cortex. Cell, 183(4), 954-967.
> * Yang, G. R., Joglekar, M. R., Song, H. F., Newsome, W. T., & Wang, X. J. (2019). Task representations in neural networks trained to perform many cognitive tasks. Nature neuroscience, 22(2), 297-306.
> * Yang & Wang, 2020: Yang, G. R., & Wang, X. J. (2020). Artificial neural networks for neuroscientists: a primer. Neuron, 107(6), 1048-1070.

---

### Official Review · Reviewer_mdrC · 2025-11-01

**Soundness:** 2
**Presentation:** 2
**Contribution:** 2
**Rating:** 2
**Confidence:** 2

**Summary:**

The paper studies how feature-specific representational dynamics shape compositional generalization in RNN models in tasks with evolving temporal structure. Using the temporally structured C-PRO task, it shows that rich learning regimes yield different temporal patterns of representational dimensionality compared to the lazy learning regimes, which correlate with better compositional generalization capability.

**Strengths:**

- The setup where tasks unfold over time and contextual information evolves is indeed underexplored in the compositional generalization literature. The paper addresses this gap.
- On the synthetic C-PRO task, the paper analyzes how rich and lazy learning regimes differ in their representation dynamics and connects these differences to their different compositional generalization capabilities.

**Weaknesses:**

- While the observed representational dynamics are empirically interesting, the paper does not explain the underlying mechanism driving such dynamics (for example, how context representations are suppressed during stimulus presentation but re-emerge during the delay period). The paper also does not clarify how these differences in representation dynamics between rich and lazy learning lead to differences in generalization capability.

- It would be helpful for the authors to discuss the related literature on rich versus lazy learning in more depth, and to clarify how their findings differ from or extend previous studies.

- It would be helpful if the authors could more explicitly highlight the main takeaway, particularly how the findings on the synthetic C-PRO task might generalize to more realistic or real-world scenarios.
- In the abstract, the authors state that prior works mainly focus on unimodal tasks. However, as I understand it, the present study also does not involve multimodal inputs. This may lead to confusion, and the authors may wish to clarify this point.

**Questions:**

- Why did the authors choose to use SGD for training instead of commonly used adaptive optimizers in deep learning such as Adam?
- Why did the authors focus on RNNs, rather than more advanced and widely used sequence models such as LSTMs or Transformers?
- Perhaps due to my limited background in cognitive science, I find certain parts of the paper hard to follow, especially:
  - The detailed setup of the C-PRO task. What are the six phases?
  - The meaning of several terms, such as neural representational dimensionality, conjunction (l190-191)
- Please also refer to the weakness.

---

> ### Author Response · Authors · 2025-11-25
> **Response to Reviewer mdrC (1/N)**
>
> > **1A (weakness): “While the observed representational dynamics are empirically interesting, the paper does not explain the underlying mechanism driving such dynamics (for example, how context representations are suppressed during stimulus presentation but re-emerge during the delay period). The paper also does not clarify how these differences in representation dynamics between rich and lazy learning lead to differences in generalization capability.”**
>
> **Reply 1A.** We thank the Reviewer for raising this weakness, as it has helped to clarify what further analyses we needed to conduct. Indeed, dimensionality of representations alone does not clarify exactly what information the representations contain at each temporal window. A more direct analysis is to quantify that information according to the features within our task design by linearly decoding those task features (i.e., rule, stimulus, and response information), which is now included (**Response 0A**). In sum, our decoding traces across task feature dimensions exhibit largely similar traces to dimensionality. This indicates that dimensionality is indeed tracking meaningful information representations across task phases. **One interesting distinction between dimensionality and decoding traces is that high dimensionality in lazy models tends to be accompanied by low feature decodability (Figure A5 E,F,H)**. This provides interpretable understanding of how the dimensionality of representation relates to information representations.
>
> * We expand on a more nuanced discussion of these claims in the caption corresponding to these results (see **Appendix Fig. A5**). We will work to integrate the take-home messages of these results into the main text in the subsequent set of revisions.
>
> * Regarding the increase in context dimensionality during the delay period: In our original task setup, the task context was injected as an input to the model throughout the entire task period. Despite this, we found that when performing context decoding analyses (**Fig. A5F**), context was not uniformly decodable throughout the entire task period. Interestingly, in lazy models, decodability of context information degraded throughout the task trial, while context dimensionality increased. In contrast, while the dimensionality of context information was overall markedly lower than in lazy models, dimensionality was positively correlated with decodability through time (**Fig. A5B vs. A5F**). These added decodability results ground our previous dimensionality results by illustrating how task information varies throughout the trial structure. In addition, we have also included an analysis using a new task variant, where task context is only presented at the initial timepoint (t=0) (results in **Fig. A4**). While we note that this task is significantly harder for models to generalize, we see similar qualitative results despite lower generalization performance.
>
> > **1B (weakness): “It would be helpful for the authors to discuss the related literature on rich versus lazy learning in more depth, and to clarify how their findings differ from or extend previous studies.”**
>
> **Reply 1B.** Prior work on studying rich and lazy learning focus primarily on learning dynamics, i.e., dynamics over training primarily in static tasks (e.g., semantic classification task, image classification, context-dependent classification) (Domine et al., 2024; Flesch et al., 2022; Chou et al., 2025). Instead, we focus on the representational dynamics of rich and lazy networks *after learning* in tasks with temporal evolution. The intention is to clarify existing narratives in the neuroscience and cognitive science literature on what representations are required and useful for generalizable behavior *during* compositional tasks. We hope this modeling work provides actionable predictions for how representations evolve through task phases in high-performing individuals. We have revised Discussion Section 4.1 accordingly, including a new section on rich and lazy learning:
>
> > “Rich and lazy learning regimes. Our work further extends the feature learning vs. function fitting trade-off in the rich/lazy spectrum (Chizat et al., 2019; Flesch et al., 2022; Tong & Pehlevan, 2025; Lippl & Stachenfeld, 2024; Liu et al., 2023). While prior work focused primarily on static tasks (e.g., see Farrell et al. (2023) for review) or the learning dynamics during optimization (Chou et al., 2025; Domine et al., 2024; Kunin et al., 2024; Liu et al., 2023), we focused on time-dependent task feature representations and their dynamics after the model has been optimized. Importantly, the emergence of the stereotyped　representational dynamics serve as a reliable predictor of generalization, and corroborated the use of efficient, intermediary contingency-based representations that enable early decision making (Ehrlich & Murray, 2022).”

---

> ### Author Response · Authors · 2025-11-25
> **Response to Reviewer mdrC (2/N)**
>
> > **1C (weakness): “It would be helpful if the authors could more explicitly highlight the main takeaway, particularly how the findings on the synthetic C-PRO task might generalize to more realistic or real-world scenarios.”**
>
> **Reply 1C**. We have now revised the Title (to **‘Task representational dynamics for compositional generalization during context-dependent cognitive tasks’**) and several Discussion sections to more clearly position this paper’s impact towards compositional tasks in neuroscience and cognitive science, rather than compositionality in RNNs in general. Indeed, while our implementation of the C-PRO task is synthetic, it is common in cognitive science and neuroscience to use highly synthetic experimental tasks for modeling in an effort to isolate the specific manipulations of interest (in this case, compositional contexts and their interaction with time-varying stimuli). This is important as it removes possible biases in naturalistic settings.  We therefore expect our work to be impactful within the confines of cognitive science and neuroscience. We have revised and refocused the Related Works sections accordingly (Section 4.1): “Representational dynamics in models for computational neuroscience”, “Rich and lazy learning regimes”, “Relation to empirical neuroscience and cognitive science”.
>
> **1D (weakness): “In the abstract, the authors state that prior works mainly focus on unimodal tasks. However, as I understand it, the present study also does not involve multimodal inputs. This may lead to confusion, and the authors may wish to clarify this point.”**
>
> **Reply 1D.** Thanks to the Reviewer for making this suggestion. The human version of the C-PRO task involves multimodal stimuli. When constructing the version of the task for computational experiments, we remapped each of these task conditions and sensory modalities to multi-hot encoded stimulus input vectors. Thus, although not considered ‘multimodal’ in a naturalistic sense, the synthetic version of the C-PRO still manipulates multi-hot encoded inputs from distinct channels (which can be considered as greatly simplified ‘modalities’). We have clarified this point in Section 2.1:
>
> > “​​We note that the multimodal stimuli in this case are different from those in a naturalistic sense. The human version of the C-PRO task involves multimodal stimuli. When constructing the version of the task for computational experiments, we remapped each of these task conditions and sensory modalities to multi-hot encoded stimulus input vectors from distinct channels, as described below.”
>
> > **1E (question): “Why did the authors choose to use SGD for training instead of commonly used adaptive optimizers in deep learning such as Adam?”**
>
> **Reply 1E.** This is a good question that we have now clarified, while incorporating an additional experiment that uses Adam in **Appendix Figure A3.** The primary reason was to maintain consistency with earlier studies that investigated learning regimes in neural networks, which established the theory and empirical experiments using vanilla SGD (Flesch et al. 2022, Chizat et al. 2019). However, in accordance with the Reviewer’s suggestion, we repeated the analyses using the Adam optimizer (**Response 0B**). Due to the adaptive learning rate in Adam, to properly observe “lazy behavior”, the norm of the initialization for lazy networks trained with Adam needed to be increased, consistent with prior work (Whitefield & Summerfield, 2025). Overall, we found similar patterns across rich and lazy learning in networks trained in Adam, particularly when observing the relationship between their representational dynamics and generalization behavior.

---

> ### Author Response · Authors · 2025-11-25
> **Response to Reviewer mdrC (3/N)**
>
> **1F (question): “Why did the authors focus on RNNs, rather than more advanced and widely used sequence models such as LSTMs or Transformers?”**
>
> **Reply 1F.** The primary focus of the current paper was to focus on understanding the relationship between the dynamics of internal representations (through time) and generalization behavior during compositional tasks, **primarily as they pertain to neuroscience and cognitive science.** This required the ability to train a distribution of models that exhibited differences in generalization behavior. We chose to train models with different learning regimes – rich and lazy – to manipulate performance and learned representations, as this has been an established manipulation and subfield within deep learning and cognitive science (Chizat et al., 2019; Liu et al. 2024; for review, see Farrell et al. 2023). For LSTMs, the role of initialization on rich/lazy learning has not yet clearly been established, which is why we opted to exclude the study of these models. As transformers are primarily a parallel sequence model (rather than a recurrent sequence model), it would have been challenging to study the evolution of representations through time, as inputs for all time points would need to be presented simultaneously. (For transformer modeling, inputs at all time points would be provided simultaneously, which would have been unnatural.)
>
> **1G (question): “Perhaps due to my limited background in cognitive science, I find certain parts of the paper hard to follow, especially:
> The detailed setup of the C-PRO task. What are the six phases?
> The meaning of several terms, such as neural representational dimensionality, conjunction (l190-191)”**
> **Reply 1G.** Thanks to the Reviewer for asking for more clarity in our writing. We edited the main text by incorporating a more detailed task description, as well as avoiding the jargon where possible.
> * Section 2.1:
> > “The temporal structure consists of six distinct task phases: 1) A task encoding phase where the 3-rule context was presented (1 timepoint, \texttt{Rule}); 2) The first stimulus presentation (2 timepoints, \texttt{Stim1\_e} and \texttt{Stim1\_l}); 3) A first delay period to measure working memory and integration of prior stimulus information and task rule information (2 timepoints, \texttt{Dly1\_e} and \texttt{Dly1\_l}); 4) The second stimulus presentation (2 timepoints, \texttt{Stim2\_e} and \texttt{Stim2\_l}); 5) A second delay period (2 timepoints, \texttt{Dly2\_e} and \texttt{Dly2\_l}); 6) The motor response window (\texttt{Resp}).”
>
> * Section 2.4 :
> > “We characterized the representational dimensionality at each time point separately. Representational dimensionality measures how many dimensions are required to capture meaningful variation in neural representations. If the 64 task contexts were stored as unique contexts (i.e., each orthogonal to each other), they would span a near-64-dimensional space. However, if some task contexts were more similar to each other (e.g., since they may have overlapping rules), these contexts could be represented in a lower-dimensional subspace where contexts are not entirely orthogonal. We quantified this using the participation ratio on the similarity matrix.:
>
>   $$ PR = \frac{(\sum_{i=1}^{s}\sigma_{i})^{2}}{\sum_{i=1}^{s}\sigma_{i}^{2}} $$
>
> where $\sigma_{i}$ represents the singular values from SVD decomposition of neural activity matrices. Higher values indicate more distributed, high-dimensional representations. We measured this metric for all the trials (global) but also in each feature (stimulus, context, response, contingency) separately by averaging the activity across the other features.”

---

> ### Author Response · Authors · 2025-11-25
> **Response to Reviewer mdrC (N/N)**
>
> **References:**
> * Chizat, L., Oyallon, E., & Bach, F. (2019). On lazy training in differentiable programming. Advances in neural information processing systems, 32.
> * Chou, C. N., Le, H., Wang, Y., & Chung, S. (2025). Feature Learning beyond the Lazy-Rich Dichotomy: Insights from Representational Geometry. arXiv preprint arXiv:2503.18114.
> * Dominé, C. C., Anguita, N., Proca, A. M., Braun, L., Kunin, D., Mediano, P. A., & Saxe, A. M. (2024). From lazy to rich: Exact learning dynamics in deep linear networks. arXiv preprint arXiv:2409.14623.
> * Farrell, M., Recanatesi, S., & Shea-Brown, E. (2023). From lazy to rich to exclusive task representations in neural networks and neural codes. Current opinion in neurobiology, 83, 102780.
> * Flesch, T., Juechems, K., Dumbalska, T., Saxe, A., & Summerfield, C. (2022). Orthogonal representations for robust context-dependent task performance in brains and neural networks. Neuron, 110(7), 1258-1270.
> * Kunin, D., Raventós, A., Dominé, C., Chen, F., Klindt, D., Saxe, A., & Ganguli, S. (2024). Get rich quick: exact solutions reveal how unbalanced initializations promote rapid feature learning. Advances in Neural Information Processing Systems, 37, 81157-81203.
> * Liu, Y. H., Baratin, A., Cornford, J., Mihalas, S., Shea-Brown, E., & Lajoie, G. (2024). How connectivity structure shapes rich and lazy learning in neural circuits. ArXiv, arXiv-2310.
> * Whitefield, M., & Summerfield, C. The Effect of Representational Compression on Flexibility Across Learning in Humans and Artificial Neural Networks. In Second Workshop on Representational Alignment at ICLR 2025.

---

### Author Response · Authors · 2025-11-25
**Overall response to all Reviewers, and summary of revisions and additional analyses (1/N)**

We thank the Reviewers for carefully assessing our manuscript, and providing many actionable suggestions. Below, we address each point, and where applicable, reference a new analysis/experiment placed in the Appendix or an in-text revision that directly addresses their concerns. For clarity, we summarize the major additional analyses here below, and reference where the Reviewers and AC can find the results in the revised PDF (changes in blue font color). Given the short turnaround, we have placed all new experiments in the Appendix, but we plan to integrate these results into the main text on the subsequent round of revisions (and if accepted, in the camera-ready version).

**0A. Temporal decoding results for all models & analyses (Appendix Fig. A5, A6; Reviewers: mdrC, zxke, xC1Z):**

* Three Reviewers asked for further analyses to provide additional insight into specifically what representational dimensionality quantified, and what kind of information was contained within those representations. **We have directly addressed these questions by providing temporal decoding analyses, decoding task feature information at each time point.** These are provided as decoding traces (across time) for sensory stimulus, task context, and motor response information for **all models (Appendix Fig. A5)**. Importantly, though we find many similarities between the decodability and dimensionality plots, decoding analyses provide direct evidence of what information exists during distinct task phases, beyond the shape (dimensionality) of the representations. Decoding analyses separating context-relevant versus irrelevant stimulus information helped clarify the functional significance of dimensionality changes across the trial (see **Appendix Fig. A6**, for stimulus dimensionality in particular). As suggested by **Reviewer zxke**, we are also in the process of including RSA-based results (i.e., comparing experimentally-designed representational similarity matrices with model representations). Preliminary findings suggest confirmatory results with the decoding analyses, but we will update the Appendix with these results when completed. We thank the Reviewers for these suggestions, as we believe that these additional analyses significantly improve the clarity and contribution of our results.


**0B. Model variations (Appendix Fig. A2, A3, A4; Reviewers: mdrC, zxke, xC1Z):**

In our Revision, we have now included experiments incorporating additional model variations. These include

* **Model size variations** (64, 128 (original), and 256 hidden units; **Reviewer xC1Z**). All main analyses (dimensionality traces and generalization accuracy) have been reproduced in these model sizes (**Appendix Fig. A2**). Importantly, **the results do not qualitatively change across these additional model variations.**

* **Task variations (Reviewers zxke, xC1Z)**. We have now reproduced all main analyses, but using a more challenging task variation where task context is only presented at t=0, rather than as a constant input throughout (**Appendix Fig. A4**). Overall, we found that fixing the task context signal to be transient at t=0 reduces overall generalizability (as it increases task difficulty, as context needs to be maintained in the model’s working memory). Nevertheless, **we still find systematic variation in the task representational dynamics (dimensionality and decoding traces) for models that generalize better** (rich initializations, ~65% generalization accuracy) than models that have poorer generalization (lazy initializations). Overall, models that generalize better exhibit representational dynamics that mimic those of the models performing the original (easier) task variation, though not exactly (likely due to significantly worse generalization behavior, i.e., 85% vs. 65%).

* Model optimization (**Reviewer mdrC**). We have also now included experimental results with models trained with Adam (original results were with vanilla SGD, due to prior theoretical results) (**Appendix Fig. A3**). Qualitatively, the additional experiments provide additional corroborating evidence of our original results. Namely, **richer representational dynamics (in terms of dimensionality and decodability) of various task features lead to improved generalization, even using the Adam optimizer.**

---

> ### Author Response · Authors · 2025-11-25
> **Overall response to all Reviewers, and summary of revisions and additional analyses (N/N)**
>
> **0C. Revised Title/abstract, additional Methods, and improved Discussion section (Reviewers: mdrC, zxke, xC1Z, o4aa):**
>
> * We have focused on explicitly positioning this manuscript’s contributions toward understanding compositionality during **context-dependent cognitive tasks** (primarily for neuroscience and cognitive science), rather than RNNs in general (**Reviewers mdrC, xC1Z, o4aa**). This is reflected in the revised Title, as well as the focus on specific Related Works sections within the Discussion (Section 4.1): “Representational dynamics in models for computational neuroscience”, “Rich and lazy learning regimes”, “Relation to empirical neuroscience and cognitive science”. We hope that this clarifies the impact within that scope.
>
> * We have revised the Discussion section to incorporate a richer discussion on rich/lazy learning, including additional references to the literature (**Reviewer mdrC, zxke**) and expanded significance of our findings (**Reviewer o4aa**). This is reflected in the new Related Works section, “Rich and lazy learning regimes” (Section 4.1)
>
> * We incorporated essential model and metric details (**Reviewer xC1Z**), and task details (**Reviewer mdrC**) in the main text (**Section 2.4**). We improved accuracy of plot descriptions (**Reviewer o4aa**) in the current version. We hope this improves the accessibility of the manuscript.
>
> **0D. Additional analyses on lazily trained models for decoding/dimensionality plots on closure and integration trials (Appendix Fig. A7; Reviewer: xC1Z):**
>
> These results provide evidence for significant variation in information content and representational dimensionality across these trial types in rich and lazy models. **These results further clarify that collapse-integration differences were more apparent and functionally significant in rich models compared to lazy models.**
>
> **0E. Additional analyses on model behavior/errors (Appendix Fig. A10; Reviewer: zxke):**
>
> We have incorporated an analysis of the generalization errors the models make to assess whether models are biased against any specific task features (task rules in particular). **Models showed no distinguishable generalization differences across task rule contexts.**
>
> **0F. Additional geometric analyses: (Appendix Fig. A8, A9; Reviewer: zxke):**
>
> We measured 1) cross-context parallelism scores (providing a similar inference to cross-condition generalization; Bernardi et al., 2020) to infer the invariance of representations across contexts (**Appendix Fig. A8**) and 2) minimum principal angle between stimulus and context subspaces to characterize orthonality/overlap between the subspaces (**Appendix Fig. A9**). **Take-home: Rich models demonstrated higher and systematic invariance (parallelism score result), and greater decoupling between stimulus and context subspaces during delay periods (subspace angle result).**
>
> In addition to the major revisions summarized here, we respond to each Reviewer’s Weaknesses and Questions point-by-point.

---

> > ### Comment · Area_Chair_2WSG · 2025-11-25
> > **Author-Reviewer Discussion**
> >
> > Dear reviewers,
> >
> > Please review the authors' response and adjust your rating accordingly. If you have any further questions, please discuss with the authors further.
> >
> > AC

---

### Author Response · Authors · 2025-12-03
**2nd overall response to all Reviewers, and summary of second revision and additional analyses**

Dear Reviewers and AC:

We have now significantly revised the manuscript a 2nd time, primarily to integrate previous experiments/results more seamlessly into the main text, while adding a new Limitations section in the Discussion. All text revisions can be tracked in the PDF (purple text = most recent revision; blue text = previous revision from 11/24/2025). Below, we highlight the main focuses of this most recent revision, which are primarily addressed to Reviewer xC1Z from the last official comment prior to the Discussion period policy change.

**0G. Integration of experiments and analyses into the main text**

Following from the extensive experiments and additions from the first revision, we have now integrated (and referenced) these results into the main text. Importantly, as we found that the new decoding results significantly improved the clarity of our earlier dimensionality results, we have now incorporated those experiments and figure panels into the main Figures (Fig. 2H-K; Fig. 4H-K). Additionally, we have included reference to each new additional figure in the Appendix (Appendix Figs. 2-8) in the main text. These can be tracked in purple text in the revised PDF.

**0H. Making the scope for cognitive science/ neuroscience application more explicit**
In our previous revision, we had significantly revised the Discussion to refocus the majority of the Discussion and related works to prior cognitive science, computational neuroscience/modeling, and empirical neuroscience literatures. In this Revision, we have additionally included a new Limitations section (colored in purple text). Finally, we have revised the Abstract and Introduction to remove text and references that could have suggested our work aimed to address compositionality in ML/RNNs broadly.

**0I. Added limitation of abstract symbolic input encodings to naturalistic stimulus processing**

Though we opted for experimental simplicity, to more explicitly investigate compositional operations, we recognize that our usage of simplistic input encodings may limit direct comparisons to naturalistic sensory processing. We have now included a new Limitations section within the Discussion to more clearly state this limitation:

> “Our study highlights the critical role of structured temporal representations in enabling robust compositional generalization in RNNs. Nonetheless, several limitations remain in the present work. First, while our implementation of the C-PRO task preserves its core cognitive operations -- compositional reasoning, working memory integration, and contingency formation -- we used symbolic input encodings (e.g., multi-hot encodings of binary features) rather than naturalistic stimuli used in the original human task design. While this limits direct comparisons to the naturalistic sensory processes involved in human cognition, this allowed us to isolate the cognitive processes of interest (compositional reasoning). Nevertheless, it will be interesting for future work to explore the temporal evolution of representations in compositional tasks that use naturalistic stimuli. Second, we studied the contribution of representational dynamics to compositional generalization by manipulating RNN weight initialization. While prior work has demonstrated that weight initialization can strongly influence generalization (Chizat al., 2019; Farrell et al., 2023), it is possible that there are other model manipulations -- as well as model architectures and optimization protocols -- that induce different relationships between representational dynamics and generalization that future work can explore. Finally, we limited our investigation to the C-PRO task. While the C-PRO task has a highly stereotyped task configuration that is commonly-used within neuroscience and cognitive science (i.e., context -> stimulus -> response), it will be important to explore compositional task designs in future work.“

---

### Note · Authors · 2026-01-28

I have read and agree with the venue's withdrawal policy on behalf of myself and my co-authors.

---

### Meta-Review · Area_Chair_6wny · 2026-01-04

**Summary:**

This paper gives a set of interesting observations about the learned task representations on a compositional generalization benchmark C-PRO. These observations can help to understand how different training regimes influence the behavior of RNN models on a compositional generational task and how the number of dimensions changes.

I noticed that the main concerns of reviewers includes:1. The scope of this paper is not very clear, who is the target audience?
2. The results are limited to a particular task and a specific model architecture.
The authors have tried to address these concerns in their responses. It is still not enough. For example, the authors only conducted experiments on RNN models. The authors argued that both LSTMs and Transformers are not good model candidates. But, Transformer is the de facto standard model. It seems that the observations from this paper may be RNN specific and not generic.

Based on this, I will not champion for this paper.

**Reviewer Concerns:**

The authors have added a set of experiments and analyses to address the questions from reviewer zxke.
A set of references are added to answer the questions from different reviewers.

However, the scope concern is still there, and it is not clear about the significance of these results for neuroscience/cognitive science.

**Reviewer Scores:**

Only reviewer xC1Z has a round of discussion with the authors. Other reviewers were not active during the rebuttal period.

---

### Decision · Program_Chairs · 2026-01-26

Reject